# FAIRNESS METRIC IMPOSSIBILITY: INVESTIGATING AND ADDRESSING CONFLICTS

## ABSTRACT

Fairness-aware ML (FairML) applications are often characterized by intricate social objectives and legal requirements, often encompassing multiple, potentially conflicting notions of fairness. Despite the well-known Impossibility Theorem of Fairness and vast theoretical research on the statistical and socio-technical trade-offs between fairness metrics, many FairML approaches still optimize for a single, user-defined fairness objective. However, this one-sided optimization can inadvertently lead to violations of other pertinent notions of fairness, resulting in adverse social consequences. In this exploratory and empirical study, we address the presence of fairness-metric conflicts by treating fairness metrics as *conflicting objectives* in a multi-objective (MO) sense. To efficiently explore multiple fairness-accuracy trade-offs and effectively balance conflicts between various fairness objectives, we introduce the *ManyFairHPO* framework, a novel many-objective (MaO) hyper-parameter optimization (HPO) approach. By enabling fairness practitioners to specify and explore complex and multiple fairness objectives, we open the door to further socio-technical research on effectively combining the complementary benefits of different notions of fairness.

## 1 INTRODUCTION

Instances of algorithmic discrimination are of growing concern in both the machine learning (ML) literature and recently, the media and greater society, due to the increasing prevalence of ML applications where *real individuals* are either directly or indirectly affected by algorithmic decisions (Angwin et al., 2016). Mirroring the complex and socio-technical nature of the machine bias problem, the research field of Fairness-aware Machine Learning (FairML) provides a collaborative space for political philosophers, social scientists, legislators, statisticians, and ML researchers. FairML has the overarching goal to define, study, detect, and mitigate algorithmic bias.

The FairML community has received widespread criticism for attempting to solve the complex, nuanced, and socio-technical problem of machine bias *algorithmically* (Hoffmann, 2019; Selbst et al., 2019). Several arguments cite that real-world applications of FairML are often categorized by a complex and unique set of social objectives and legal requirements (B.Ruf and Detyniecki, 2021). Due to their complexity, these criteria are unlikely to be captured by single coarsely-grained statistical measures of fairness. In such instances, FairML methods that incorporate only a single notion of fairness may return a so-called *fair* model that satisfies one notion of fairness by violating another potentially relevant one - ultimately resulting in negative social consequences. In the presence of what we call fairness metric *conflicts*, we show that these one-sided FairML approaches are of limited utility to real-world practitioners, and even risk providing a false sense of security and trust.

Rather than resisting these criticisms, we embrace them, agreeing that FairML approaches that over-simplify the complex and socio-technical nature of the FairML problem actually risk doing more social harm than good. In recent years, the topic of fairness has gained popularity in the Automated Machine Learning (AutoML) literature (Wu and Wang, 2021; Perrone et al., 2021; Schmucker et al., 2020). Fairness-aware AutoML approaches typically formulate fairness as a multi-objective (MO) hyperparameter optimization (HPO) problem, varying common ML design decisions (tree height, number of neural network layers, etc.) in order to search for a Pareto Front of fair and accurate ML models (Weerts et al., 2023). Fairness-aware AutoML holds key advantages in transparency, not

only providing the practitioner with a Pareto Front of fair and accurate ML models to choose from, but insight into the problem-specific objective landscape through MO analysis.

In order to bridge this gap, we formulate FairML as a MaO problem and critically evaluate the motivation and merit of simultaneously optimizing for multiple user-defined notions of fairness. After summarizing related work (Section 2) and describing background on FairML, MOO and, HPO (Section 3), we make the following contributions in our methodology and experiments (Sections 4 and 6):

1. We present an empirical study regarding the presence and nature of fairness metric conflicts. On several real-world FairML problems, we show that optimizing for particular notions of fairness may inadvertently violate other potentially relevant ones. Fairness metrics can and should be treated as potentially conflicting objectives in an MO sense.

2. We propose a start-to-finish Many-Objective framework, coined ManyFairHPO, that enables fairness practitioners to specify and define multiple conflicting fairness objectives, with the overarching goal of balancing fairness objectives and mitigating conflict-related risk.

## 2 RELATED WORK

The characterization of fairness as an inherently MO problem has received significant attention in the literature (Martinez et al. (2020), Islam et al. (2021)), with several works citing the importance of considering trade-offs in the FairML problem landscape. In this section, we discuss the critical issue of optimization problem formulation for fairness, ultimately motivating our Many-Objective (MaO) approach.

We begin by discussing several works that formulate the FairML task as a Constrained Optimization (CO) problem. In particular, Perrone et al. (2021) expands upon common bias-mitigation techniques (Section A.3) to integrate constraints across multiple fairness metrics. Relatedly, Hsu et al. (2022) employ Mixed Integer Linear Programming (MILP) to enforce constraints across various fairness criteria. Their primary objective is to explore and push the boundaries set by the Impossibility Theorem while placing a strong emphasis on interpretability. While the above-stated works make relevant contributions they also open up the door to the following discussions regarding which optimization problem formulation is best suited to the FairML task (Weerts et al., 2023).

First, although previous socio-technical work (B.Ruf and Detyniecki, 2021) provides problem-specific guidelines for fairness metric selection, the range of achievable fairness metric values (especially in the presence of conflicts) is highly data and model-dependent, making the setting of realistic fairness constraints an inherently non-trivial task (Weerts et al., 2023). While we align with the argument of Perrone et al. (2021) that CO holds computational efficiency advantages in cases where fairness objectives and reasonable constraints are clearly defined, we argue that this is a special and typically unrealistic scenario, especially when constraining across multiple, potentially conflicting notions of fairness (Section 6.1). The MO problem formulation, on the other hand, offers increased flexibility, as it requires little prior knowledge regarding the fairness objective landscape (Weerts et al., 2023).

Second, while CO approaches hold advantages in computational efficiency by focusing the search on the constrained region of interest, they typically do not explore the entire Pareto efficient frontier of all objectives Weerts et al. (2023), often leaving quality (either in terms of performance or fairness) on the table. By exploring the Pareto Front(s), MO approaches also provide interpretable insights into the overall objective landscape, a key aspect in building FairML trust and aiding practitioners in the iterative FairML design cycle (Weerts et al., 2023). Another key advantage of the MO problem formulation is its natural extension to the MaO case, which incorporates multiple user-defined fairness metrics as additional objectives. Although Schmucker et al. (2020) argues that their methodology is extensible to the MaO case, they do not test this hypothesis or explore its benefits. In this study, we fill this gap in the FairML literature, first justifying the treatment of fairness metrics as potentially conflicting objectives (Section 6.1) and then showing that MaO is capable of not only the exploration of multiple fairness-accuracy trade-offs but also of balancing fairness metric conflicts (Section 6.2).

## 3 BACKGROUND

### 3.1 FAIRNESS-AWARE MACHINE LEARNING

In this section, we provide an introduction to fairness, its many definitions, bias mitigation strategies, and the theoretical trade-offs between fairness and performance objectives. Machine bias occurs when an ML algorithm learns to make discriminatory predictions with respect to legally protected or ethically sensitive attributes. FairML approaches seek to detect and mitigate machine bias at various stages of the ML pipeline and can be effectively divided into pre-processing, in-processing, and post-processing techniques, which detect and mitigate bias in either 1) the input data, 2) the training algorithm or 3) the model's predictions (Pessach and Shmueli, 2022). In recent years, the objective of fairness has gained increased popularity in the AutoML community, resulting in several studies that achieved competitive performance to specialized bias-mitigation techniques by simply varying the hyperparameters of ML models (Perrone et al., 2021; Schmucker et al., 2020; Dooley et al., 2022). For an in-depth discussion on fairness-aware AutoML, we refer to Weerts et al. (2023).

Despite the strong performance of bias-mitigation techniques and fairness-aware AutoML, a lingering question in the fairness literature remains an introspective one: how do we define fairness itself? Grappling with an already nuanced and philosophical question (Binns, 2018), as well as a growing set of social objectives and legal requirements, the FairML community has proposed over 20 different fairness metrics (Barocas et al., 2019), which seek to quantify the bias or fairness of a predictive algorithm. The FairML problem is formally defined below.

**Definition (FairML Problem)** Given a data set $\mathcal{D} = (X, Y, A)$ of $n$ features and $m$ samples $X \in \mathbb{R}^{m \times n}$, a binary target $Y \in \{0, 1\}^m$, and a binary protected attribute $A \in \{0, 1\}^m$, find a model $\mathcal{M} : X \to \hat{Y}$ that is fair with respect to $A$ at either the population or individual level.

The three main notions of group fairness are Independence, Separation, and Sufficiency (Barocas et al., 2019). These notions correspond closely to the group fairness metrics Statistical Parity (DDSP), Equalized Odds (DEOD), and Equal Opportunity (DEOP). While group fairness and its many metrics operate upon the *egalitarian principle* that positive outcomes should be distributed equally across society, individual fairness is based on the notion of *just deserts* (Binns, 2018). Associated metrics include Inverse Distance (INVD) or similarity-based fairness metrics, and stipulate that similar individuals (based on a set of *legitimate factors*) receive similar outcomes (Barocas et al., 2019). We note that defining a set of legitimate factors that is truly independent of the protected attribute is an inherently challenging task. In this study, we opt for a simplified definition of INVD, which treats the labels as a legitimate factor for admissions, as proposed by Berk et al. (2017). We provide a formal definition of the above-mentioned fairness metrics in Table 2 (in the appendix).

A central concept to this study in particular is the *Impossibility Theorem of Fairness*, which states that the three main notions of group fairness (Independence, Separation, and Sufficiency) cannot all be satisfied at once (Miconi, 2017). According to Barocas et al. (2019)

> "What this shows is that we cannot impose multiple criteria as hard constraints.
> This leaves open the possibility that meaningful trade-offs between these different
> criteria exist."

### 3.2 MULTI-OBJECTIVE OPTIMIZATION

MOO is an increasingly relevant topic in ML research, as ML applications are increasingly scrutinized with respect to environmental, logistical, and social requirements (Karl et al., 2022). MOO seeks to provide solutions to an optimization problem under a set of two or more *conflicting* objectives (Sharma and Kumar, 2022). Objectives are called conflicting when gains in one objective trade-off against losses in the other.

Let $\Lambda$ be a design space representing all possible design decisions. The goal of MOO is to find a set of design solutions $\lambda \in \Lambda$ that minimize a multi-criteria objective function $f : \Lambda \to \mathbb{R}^d$, which returns a vector of costs $\langle f_1(\lambda), f_2(\lambda), ..., f_d(\lambda) \rangle$ with respect to each objective. We consider the strict partial Pareto order $\prec_{\text{Pareto}}$ where $x \prec_{\text{Pareto}} y$ iff. $x_i \leq y_i$ for all $1 \leq i \leq d$ and $x_j < y_j$ for at

least one $j$. If we consider all possible values $\mathcal{Y} = f(\Lambda)$ we obtain a set of minimal values:

$$\mathcal{P}(\mathcal{Y}) := \{y \in \mathcal{Y} : \{y' \in \mathcal{Y}, y' \prec_{\text{Pareto}} y, y' \neq y\} = \emptyset\} \tag{1}$$

called the *Pareto Front*. Any value in $\mathcal{P}(\mathcal{Y})$ is called *non-dominated*, i.e., there is no other solution that is as least as good as this value but strictly better in at least one of the objectives.

The *hypervolume* is a quality indicator and represents the volume of the dominated region of the objective space with respect to a reference point $r$.

$$\mathcal{H}(\mathcal{P}) := \text{Vol}(\mathcal{P}(\mathcal{Y})) = \text{Vol}(\{y \in \mathcal{Y} : \{y' \in \mathcal{Y}, y' \prec_{\text{Pareto}} y, y' \neq y\} = \emptyset\}) \tag{2}$$

In this study, we focus on *normalized hypervolume*, an extension of the concept of hypervolume which scales the empirical value to a (0, 1) range, making it comparable across different optimization tasks. For a further discussion on normalized hypervolume and other MO quality indicators, we refer to (Hansen et al., 2022). Many Objective (MaO) optimization is the extension of MOO to a set of three or more objectives. MaO algorithms are conceptually similar to MOO algorithms but often implement additional quality-diversity measures to ensure sufficient exploration of all objectives (Deb and Jain, 2013).

### 3.3 HYPERPARAMETER OPTIMIZATION

Hyperparameter optimization (HPO) seeks to automate the trial-and-error process of designing and deploying ML models (Bischl et al., 2023). HPO has been shown to consistently improve performance across a wide variety of ML tasks and has since become a crucial aspect of the ML design cycle. Another advantage of HPO is its extension to the MO case, where varying ML hyperparameters can have an impact on not only performance, but other ML objectives, such as interpretability, energy efficiency, and namely, fairness (Karl et al., 2022). In the context of fairness-aware AutoML, recent studies have shown a significant impact of regularizing hyperparameters on commonly used fairness metrics. Fairness-aware HPO provides a convenient framework for the FairML problem and is also extensible to include preprocessing, in-processing, and post-processing techniques (as well as their own hyperparameters) as part of the search (Wu and Wang, 2021).

## 4 FAIRNESS AWARE MANY-OBJECTIVE OPTIMIZATION: MANYFAIRHPO

In this section, we describe our MaO approach to fairness, ManyFairHPO, in which we provide a start-to-finish framework for MaO optimization and model selection for FairML problems characterized by multiple, potentially conflicting fairness objectives. We also provided a visual overview of ManyFairHPO in Figure 1.

### 4.0.1 FAIRNESS METRIC SELECTION AND RISK IDENTIFICATION

ManyFairHPO begins with a domain-knowledge-driven deliberation as visualized in Figure 1, following the guidelines of B.Ruf and Detyniecki (2021) in order to determine a set of fairness metrics that effectively capture different aspects of the various social objectives and requirements of the FairML problem at hand[1]. Another crucial step (unique to the ManyFairHPO framework) is the identification of fairness metric conflicts that pose downstream risks. For example, the conflict between statistical parity (DDSP) and Equalized Opportunity or individual fairness (INVD) has been shown to cause self-fulfilling prophecy Dwork et al. (2011), specifically when demographic quotas are filled by carelessly selecting underprivileged candidates or applicants. The requirements at this stage of ManyFairHPO are 1) a set of performance and fairness metrics $\{f_0, f_1, f_2, ...\}$ and 2) a set of weights $\langle w_0, w_1, ... \rangle$ that represent our metrics preferences, and identification of conflict-related risks with respect to pairs of fairness metrics $(f_i, f_j)$ and where they might occur in the objective space.

### 4.1 MANY-OBJECTIVE OPTIMIZATION

Given the contrasting nature of fairness objectives, we propose a MaO-HPO problem formulation for fairness, which takes as objectives a performance metric $f_0$ and two or more notions of fairness

---

[1] As in any ML task, the choice of performance metric is equally critical, and should also consider problem-specific characteristics.

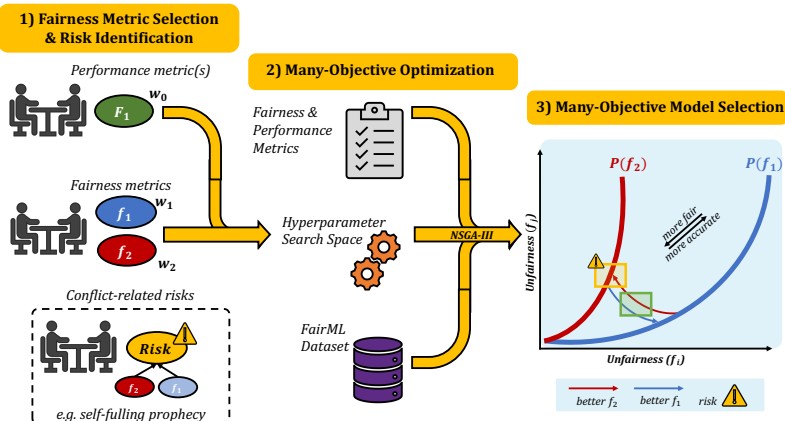

Figure 1: **ManyFairHPO**: A many-objective, fairness-aware hyperparameter optimization framework designed to efficiently explore multiple fairness-accuracy trade-offs, as well as balance conflicts between multiple contrasting notions of fairness.

$\{f_1, f_2, ...\}$ (Figure 1). We apply the popular MaO algorithm NSGA-III Deb and Jain (2013) in order to efficiently explore the objective space of fairness-accuracy trade-offs and fairness metric conflicts, resulting in a MaO Pareto Front $P(\mathcal{Y}_{MULTI})$. For a more in depth discussion of NSGA-III and Evolutionary Algorithms in general we refer to Appendix B.

## 4.2 MANY-OBJECTIVE MODEL SELECTION

As visualized in Figure 1, the final stage of ManyFairHPO involves selecting a model that effectively captures social objectives and mitigates conflict-related risk. In order to do so, we propose a single objective scalarization approach inspired by Cruz et al. (2021) that incorporates user preferences towards performance and fairness metrics as weights in a linear combination. Such weights should capture the relative importance of metrics in the problem at hand and crucially, guide the model selection away from regions of the objective space where fairness metric conflict-related risks occur.

Given a MaO Pareto Front $P(\mathcal{Y}_{MULTI})$ where each solution $\lambda$ contains a vector of performance and fairness metric values $\langle f_0(\lambda), f_1(\lambda), ...\rangle$, and a vector of weights $\langle w_0, w_1, ...\rangle$, the MaO model selection task is simplified to the following:

$$\lambda^* = argmin_\lambda \sum w_i \cdot f_i(\lambda) \tag{3}$$

For the sake of simplicity, we define Equation 3 as a simple linear combination. However, more complex (e.g. exponential) functions can also be used to incorporate the utility and marginal gains of metrics. In addition, we propose the mitigation of conflict-related risks via the weighing of fairness metrics that draw the model selection away from these regions. In order to make the effect of objective weights on conflict-related-risk avoidance more concrete, we provide a simple example in Figure 1. Given a risk that occurs when fairness metric $f_2$ is satisfied and $f_1$ is not (red box), we assign a weight $w_2$ to the second fairness metric, pulling the model selection away from the risk region (green box). We highlight, however, that risk regions might be more rigid and complex, and recommend further exploration in future works regarding the optimal avoidance of these regions.

## 5 EXPERIMENTAL SCOPE

Our experimental scope spans over Random Forest (RF), XGBoost (XGB) and Multi-Layer Perceptron (NN) Pedregosa et al. (2011) and their HPOBench hyperparameter search spaces (Table 1) and

the Bank Marketing[2], German Credit, Adult Census Income, COMPAS Criminal Recidivism, and Lawschool Admissions data sets Dua and Graff (2017) (Table 4). The fairness metrics within our experimental scope are drawn from IBM's `aif360` library, which provides fairness metrics, bias mitigation techniques (we provide a comparison with Exponentiated Gradient Reduction (EGR) in Appendix A) and a scikit-learn compatible interface for fairness research and real-world FairML applications of FairML (Bellamy et al., 2018). In order to align with the FairML literature, we select the most commonly used fairness metrics for our analysis: DDSP, DEOD, DEOP, and INVD (Appendix Table 2). We highlight however, that in real-world applications of ManyFairHPO, practitioners are not limited to these fairness objectives and are encouraged to hand-select metrics that effectively capture problem-specific social objectives and requirements (Section 4.0.1. In order to encourage reproducibility and transparency, we provide all source code for experiments and analysis at https://anonymous.4open.science/r/ManyFairHPO-ICLR.

In order to simulate standard bi-objective optimization, we first apply NSGA-II to optimize for $F_1$-Score and a single fairness metric $f_i$, leading to several two-dimensional fairness-accuracy Pareto Fronts that minimize the multi-criteria objective function described in Appendix B. For each hyperparameter configuration evaluated during bi-objective experiments, we also record the values of non-optimized fairness metrics for later analysis. We run bi-objective optimization over all three HPO search spaces, five FairML datasets, and four fairness metrics, resulting in a total of 60 bi-objective experiments, each of which is repeated 10 times using different random seeds (600 total runs). We apply MaO optimization to optimize for a five-dimensional Pareto Front $P(\mathcal{Y}_{MULTI})$ of $F_1$-Score and all four fairness metrics from our bi-objective experiments (Appendix Table 2). We apply ManyFairHPO over all models and data sets in our experimental scope, resulting in a total of 15 additional MaO experiments, also repeated using 10 random seeds (150 total runs). An overview of our experimental scope is provided in Appendix Table 3.

In the following research questions and case study, we critically evaluate the need for FairML approaches like ManyFairHPO that incorporate and optimize for multiple user-defined notions of fairness.

**RQ 1**: *Fairness Metric Conflicts* - To what extent do the fairness metric conflicts implied in the theoretical literature occur in practice? Which problems do they occur on, and what are their societal implications?

**RQ 2**: *Fairness-aware Many-Objective Optimization* - Given the potentially conflicting nature of fairness metrics, what is the effect of ManyFairHPO in simultaneously optimizing for multiple notions of fairness?

**Case Study**: *Lawschool Admissions* - We delve into a University Admissions scenario as an exemplary case in which ManyFairHPO can be applied. Relating social objectives and downstream risks to a set of multiple fairness metrics, we simulate the MaO problem formulation and ultimately model selection that incorporates multiple conflicting performance and fairness objectives.

## 6 RESULTS

In this section, we carry out several specifically designed experiments in order to address our research questions regarding the identification and resolution of fairness metric conflicts.

### 6.1 RQ1: FAIRNESS METRIC CONFLICTS

Our first research question aims to evaluate the degree to which the fairness metric conflicts implied in the theoretical literature occur in practice, in order to build an empirical basis for the treatment of fairness metrics as conflicting objectives. In order to address this question, we calculate the contrast between fairness metrics $C(f_i, f_j)$ as described in Definition B.1.

First, we calculate $C(f_i, f_j)$ over all data sets, HPO search spaces, and pairs of fairness metrics and provide a summary of observed conflicts in Figure 2. A light red cell in row $j$, column $i$, indicates that optimizing for $F_1$-Score and fairness metric $f_i$ fails to optimize for $F_1$-Score and fairness

---

[2]For the Bank Marketing data set, the protected attribute "age" is binarized to 0 if less than the median value (25), and otherwise to 1.

metric $f_j$. A blue cell indicates a *negative* conflict, where optimizing for $F_1$-Score and another fairness metric discovers *stronger* solutions than optimizing for a fairness metric directly. This result suggests a potential interaction between fairness metrics and is further explored in Appendix A.1.

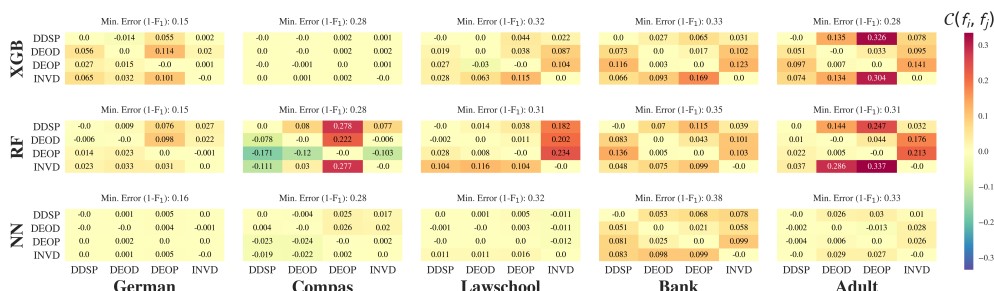

Figure 2: **Fairness Metric Conflicts**: Overview of fairness metric conflicts, measured by the degree to which optimizing for one fairness-accuracy trade-off optimizes for another. A dark red cell indicates a severe conflict. Fairness metric conflicts occur to different extents on different HPO search spaces and data sets.

Having identified the data-dependant prevalence of fairnerss metric conflicts, we direct our attention to a conflict on the Lawschool Admissions data set, which provides a highly contextual FairML problem setting, especially considering the current debate regarding Affirmative Action in University admissions (Santoro, 2023). On the RF-Lawschool experiment, we observe a conflict in Figure 2 (middle cell) between individual fairness metric INVD with respect to group fairness metrics DEOP ($C = 0.234$), DEOD ($C = 0.202$) and DDSP ($C = 0.182$). This conflict is explained in Appendix Figure 10, which shows that $\mathcal{P}(\mathcal{Y}_{INVD})$ fails to approximate group fairness Pareto Fronts, especially towards the high error, low group unfairness regions of the $(F_1, DDSP)$ and $(F_1, DEOD/P)$ objective spaces. In order to understand the social implications of this particular fairness metric conflict, we examine the predictive behavior of two models: one selected from the high error, low individual unfairness extreme point of $\mathcal{P}(\mathcal{Y}_{INVD})$, and another selected at a similar error level from $\mathcal{P}(\mathcal{Y}_{INVD})$. We provide a detailed summary of the predictive behavior of these models in Appendix Figure 6. We observe that both models exhibit relatively low group and individual unfairness (but high error) by increasing the predicted qualification rate for Black students from $P(Qualified|Black) = 0.01$ in the data to $P(Accept|Black) = 0.08$ as per the individually fair model and to $P(Accept|Black) = 0.13$ as per the group-fair model.[3]

The model that prioritizes individual fairness rejects 3% more qualified and 2% more unqualified Black applicants than the group-fair model. This leads to an improvement in INVD, as 2% fewer similarly unqualified Black and White applicants receive different outcomes. However, the group-fair model accepts 5% more Black applicants compared to the individually-fair model. This results in a reduction of the disparity between group acceptance rates (DDSP) from 23% in the individually-fair model to 18% in the group-fair model. It is important to note that although the group-fair model achieves a more diverse accepted class, it does so by accepting 2% more unqualified Black applicants than the individually-fair model. Such an admissions strategy can lead to a *self-fulfilling prophecy* as described by Dwork et al. (2011) if the accepted underprivileged students do not succeed in their studies.

We thus confirm that optimizing for a particular fairness metric does not imply optimizing for others and that MO exploration of fairness metric conflicts can provide interpretable insight into problem-specific objective landscapes. We also provide a contextual example of the potential negative social consequences of fairness metric conflicts, outlining the social importance of considering fairness metric conflicts and setting fairness objectives that span across multiple notions of fairness. The presence and potential negative social implications of these conflicts guides us toward the treatment of fairness metrics as potentially conflicting objectives.

---

[3]We note that these percentages certainly do not reflect the true qualification rate of Black applicants, and is likely influenced by underrepresentation in the data, self-fulfilling prophecies, and increased challenges for Black accepted students.

## 6.2 RQ2: FAIRNESS-AWARE MANY-OBJECTIVE OPTIMIZATION

Recognizing the impact of fairness metric conflicts being specific to the FairML problem at hand, we systematically assess the effectiveness of optimizing for multiple fairness metrics, a proposition introduced but not comprehensively investigated by Schmucker et al. (2020).

As an initial validation step, we first verify that optimizing for multiple fairness metrics together is at least as effective as optimizing for the same metrics separately. To empirically validate this hypothesis, we plot the progression of hypervolume regret (Appendix Figure 11) between MO and MaO experiments, defined as the loss in $\mathcal{H}_{f_i}$ from optimizing for all fairness metrics $f_{1:d}$ together (Appendix B). We observe that the majority of MaO experiments converge to zero regret, indicating that similar fairness-accuracy Pareto Fronts were achieved in MaO and bi-objective experiments.

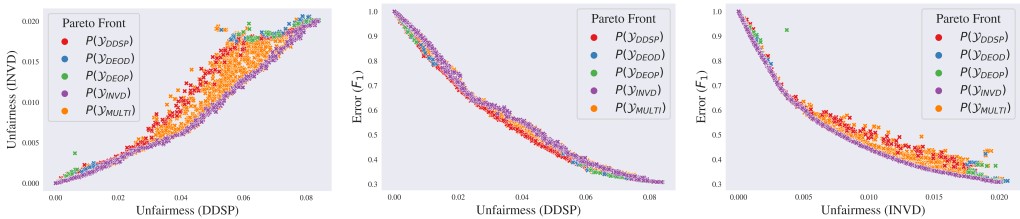

Figure 3: **Conflict Resolution (RF-Adult)**: Visualization of the MaO Pareto Front in the presence of a fairness metric conflict between INVD and DDSP (left) and a simulation of ManyFairHPO model selection (right). ManyFairHPO is capable of balancing fairness metric conflicts.

We also delve into the effectiveness of MaO in achieving a balance between fairness metrics conflicts, returning our focus to a conflict observed between INVD and DDSP ($C = 0.074$) from the RF-Adult experiment (Figure 2 top-right). The Pareto Fronts $\mathcal{P}(\mathcal{Y}_{DDSP})$ and $\mathcal{P}(\mathcal{Y}_{INVD})$ leave a noticeable gap in the $(INVD, DDSP)$ objective space (Figure 3 left), failing to identify solutions that strike a balance between individual (INVD) and group unfairness (DDSP). In contrast, the many-dimensional Pareto Front $\mathcal{P}(\mathcal{Y}_{MULTI})$ includes solutions that encompass this trade-off, granting practitioners a wide range of choices when it comes to selecting a model. We observe this phenomenon across all experiments (Appendix Figure 13), confirming that the MaO Pareto Front contains solutions that trade-off fairness metric conflicts.

## 6.3 CASE STUDY: LAWSCHOOL ADMISSIONS

We now delve into the Lawschool Admissions dataset as an exemplary scenario in which Many-FairHPO can be applied, with the goal of grounding our technical solution to real-world FairML applications. We also provide clear instructions on how to navigate MaO model selection in the face of fairness metric conflicts and their associated risks. University Admissions is a topical scenario where different notions of justice are commonly debated in the context of rule-based systems or human admissions committees. Affirmative Action, until it was overruled by the US Supreme Court earlier this year Santoro et al. (2022), was a policy designed to increase admissions opportunities for underprivileged or underrepresented racial groups. Controversially, Affirmative Action allowed for so-called positive discrimination in order to encourage racial diversity, with the overarching goal of using education as a means to correct complex systems of racial inequality. Such a strategy sparks counterarguments related to individual injustice, as equally qualified applicants from different racial groups might receive unequal treatment, making it harder for potentially qualified applicants from the privileged group to receive an acceptance. As described in Section 6.1, a less political counterargument to positive discrimination cites the risk of *self-fulfilling* prophecy, which occurs when underprivileged students are carelessly selected for admissions.

In the context of the Lawschool Admissions dataset, we relate the aforementioned political arguments to group and individual fairness metrics: DDSP and INVD. We also define the downstream risk of self-fulfilling prophecy which occurs when DDSP is satisfied and INVD is not. Given the characteristics of the problem discussed above, we define the weights $\mathbf{w} = \langle 0.5, 0.3, 0.2 \rangle$ to reflect the importance of performance and fairness metrics $F_1$-score, DDSP, and INVD respectively. Performance is given the highest weight $w_0 = 0.5$ in order to prioritize a well-fit classifier, DDSP is set

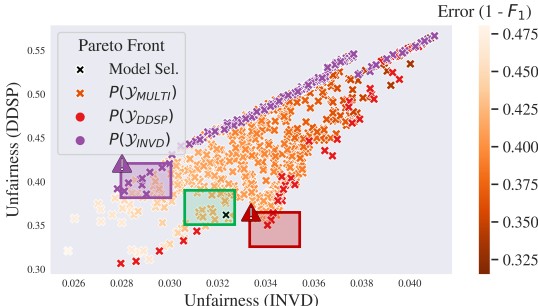

Figure 4: **Many-Objective Model Selection**: Simulated MaO model selection on the Lawschool Admissions dataset in the face of a conflict between group and individual fairness metrics DDSP and INVD. ManyFairHPO incorporates preferences towards performance and fairness metrics to balance fairness metric conflicts and avoid conflict-related risks.

to second priority ($w_1 = 0.3$) to encourage the diversity in the accepted class, and INVD is given $w_2 = 0.2$ in order to avoid self-fulling prophecy, ensuring that when positive discrimination occurs, underprivileged students are carefully selected.

Our priority towards accuracy leads us to a moderately accurate, ($F_1 \approxeq 0.5$) moderately fair region of the $(INVD, DDSP)$ objective space. In Figure 4 we visualize and evaluate three possible model selection decisions. The purple box contains models near the $P(\mathcal{Y}_{INVD})$ Pareto Front, which sacrifices diversity for individual fairness ($DDSP > INVD$). Due to our preference for diversity, we may alternatively prefer to select from the red box. Although the red box captures our diversity objective, it significantly sacrifices individual fairness, an admissions strategy shown in Section 6.1 to increase the acceptance likelihood for underqualified members of the underprivileged group. Due to the risk of creating a self-fulfilling prophecy, which could lead to further future discrimination, we select a model from the green box (black cross). This model selection (determined by our proposed weights) effectively captures our preferences toward diversity and successfully avoids regions of the objective space associated with self-fulfilling prophecy. We thus show that the ManyFairHPO framework provides a start-to-finish, domain-expert in-the-loop approach for fair decision-making in FairML problems categorized by multiple fairness objectives and conflict-related risks.

## 7 CONCLUSION

In this paper, we introduce a Many-Objective framework for fairness coined ManyFairHPO, which enables practitioners to specify and optimize for multiple, potentially conflicting notions of fairness. The ManyFairHPO framework begins with a domain-knowledge driven deliberation to assign objective weights towards fairness metrics and identify regions of the fairness objective space that correspond to downstream conflict-related risks. After performing Many-Objective Optimization to thoroughly explore the Pareto Front of performance and multiple fairness objectives, we propose a model selection strategy that incorporates domain-expert preferences towards fairness metrics to balance objectives and mitigiate risk. In our experimental results, we also provide a Lawschool Admissions case study exemplifying how social objectives and risks can be mapped to fairness objectives and weights, ultimately resulting in a model selection that effectively balances individual justice with diversity objectives while mitigating the risk of self-fulfilling prophecy.

The results of this study lead to several key recommendations for the FairML community. Firstly, it is essential to recognize and treat fairness metrics as potentially conflicting objectives. Acknowledging the existence and social consequences of fairness metric conflicts, FairML approaches should be capable of incorporating and balancing multiple fairness objectives. ManyFairHPO opens the door for socio-technical discussions of how fairness metrics can be used together in balance to meet complex social objectives and requirements, shifting the debate from "Which fairness metric is appropriate for a given problem?" to "Which balance of fairness metrics would optimally achieve our social objectives?" Such discussions could lead to guidelines on which sets of fairness metrics are relevant to certain problem types and which conflicts to be aware of.

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
