# A SUPPLEMENTARY RESULTS

## A.1 CRIMINAL RECIDIVISM

The Compas data set is a seminal instance of algorithmic bias, where a sentencing algorithm used in the Florida judicial system to predict criminal recidivism (likelihood of re-offending) to aid parole decisions was found to be severely biased against Black defendants (Angwin et al., 2016). In this section, we take a closer look into the so-called *negative conflicts* we observed in the RF-Compas experiment, where optimizing for different fairness metrics found stronger solutions than optimizing for a fairness metric directly. This result suggests a possible explanation for the strong performance of the MaO problem formulation in this scenario (Figure 11), where interaction between fairness metrics enables MaO to discover strong overall solutions.

In Figure 2 we observe negative contrast values between DDSP with respect to DEOD ($C = -0.078$), DEOP ($C = -0.171$), and INVD ($C = -0.111$), indicating that a fairer solutions in terms of DEOD/P and INVD wer discovered when optimizing for $F_1$-Score and DDSP. We also observe MaO experiments in Figure (11) with negative regret (-5% to -10%), indicating that a higher $\mathcal{H}_{DEOD/P}$ and $\mathcal{H}_{INVD}$ was achieved by the MaO experiment than by their corresponding bi-objective experiments. These results are attributed to a single model discovered on $\mathcal{P}(\mathcal{Y}_{DDSP})$ which achieves reasonable accuracy ($1 - F_1 = 0.3$) and the lowest unfairness in terms of all fairness metrics (Figure 5).

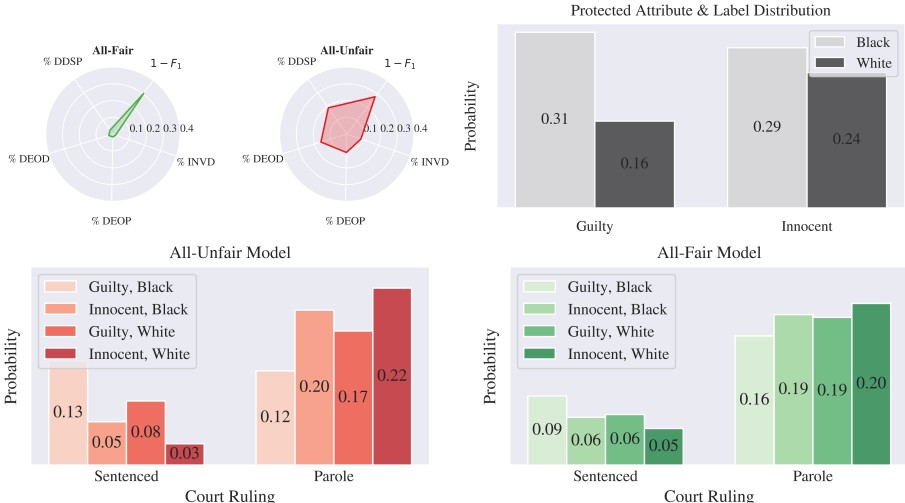

Figure 5: **Negative Conflicts (RF-Compas)**: Overall-fair model discovered in the RF-Compas experiment, which minimizes all fairness metrics at a reasonable accuracy level. The All-Fair model increases the parole rate for Black defendants that did in fact re-offend to the same rate as White applicants, resulting in low group and individual unfairness.

In order to better understand how strong overall fairness was achieved by this model, we compare its behavior with a slightly more accurate ($1 - F_1 = 0.27$) but less fair model in terms of all fairness metrics (Figure 5). In comparison, the all-fair model has a lower sentencing rate for Black defendants that re-offended ($P(Sentenced|Black, Guilty) = 0.09$) than the overall-unfair model ($P(Sentenced|Black, Guilty) = 0.13$). However, because both models have a high parole rate for White defendants that re-offend ($P(Parole|White, Guilty) \geq 0.20$), the decreased sentencing rate from the overall-fair model has the effect of improving overall fairness. First of all, the between-group parole rate $P(Parole|White) - P(Parole|Black)$ is improved from DDSP = 0.07 in the overall-unfair model to DDSP = 0.04 in the overall-fair model. In addition, the between-group parole rate for non-re-offending defendants $P(Parole|White, Innocent) - P(Parole|Black, Innocent)$ is improved from DEOP = 0.02 in the overall-unfair model to DEOP = 0.01 in the overall-fair model. Finally, similarity-based individual fairness INVD is also improved in the overall-fair model, as 4% more similarly re-offending defendants receive similar parole outcomes. This result suggests that

optimizing for multiple notions of fairness can have the effect of unlocking regions of the objective space that are otherwise inaccessible using the MO problem formulation.

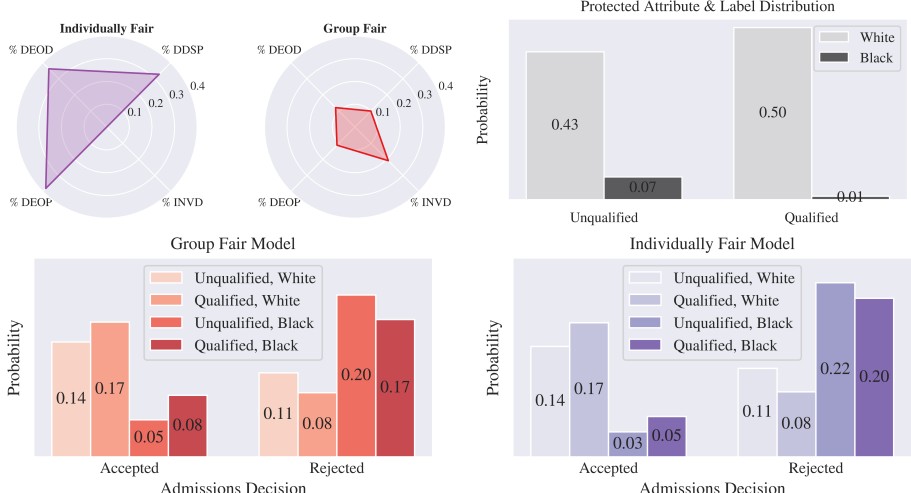

Figure 6: **Conflict Implications (RF-Lawschool)**: Overview of the conflict between individually-fair and group-fair models discovered in the RF-Lawschool experiments. The individually fair model has a higher rejection rate, resulting in fewer similarly unqualified Black and White students receiving different outcomes (individual fairness). However, this strategy also increases between-group acceptance rates (group unfairness).

## A.2 CONFLICT EXPLANATION

In this section, we direct our focus to the RF-Adult experiment (Figure 2 middle-right), which indicates a severe, moderate, and weak conflict between DDSP with respect to DEOP ($C = 0.326$), DEOD ($C = 0.135$), and INVD ($C = 0.078$). To explain these conflicts, we provide the normalized hypervolume regret $R_{MO}$ of fairness-accuracy Pareto Fronts (Figure 7), which describes the loss in $\mathcal{H}_{f_i}$ from optimizing for another fairness metric $f_j$. For a formal definition of $R_{MO}$, we refer to Appendix B.

For example, with respect to DDSP (Figure 7 top-left), we observe that $\mathcal{P}(\mathcal{Y}_{DEOP})$ converges to the worst $\mathcal{H}_{DDSP}$ (regret $R_{MO} = 0.35$), while $\mathcal{P}(\mathcal{Y}_{DEOD})$ achieves a better, yet still suboptimal $\mathcal{H}_{DDSP}$ (regret $R_{MO} = 0.2$). Through visualizing the $(F_1, DDSP)$ objective space locations of these Pareto Fronts (Figure 7, bottom left), we observe that optimizing for $F_1$-Score and DEOD/P only discovers solutions near the extreme points of $\mathcal{P}(\mathcal{Y}_{DEOP})$. In that figure, we also observe that $\mathcal{P}(\mathcal{Y}_{INVD})$ achieves a nearly optimal $\mathcal{H}_{DDSP}$ (regret $R_{MO} = 0.05$), i.e., optimizing for $F_1$-Score and INVD closely approximates $\mathcal{P}(\mathcal{Y}_{DDSP})$ (Figure 7 top-left).

## A.3 COMPARISON TO BIAS-MITIGATION

In this section, we compare our Pareto Fronts obtained from ManyFairHPO to the state-of-the-art bias mitigation technique Exponential Gradient Reduction (EGR), which proposes a reduction of the FairML task to a series of cost-sensitive classification problems Agarwal et al. (2018). In order to provide an apples-to-apples comparison with our MaO Pareto Fronts, we post-process the most accurate, least fair hyperparameter configurations with EGR for 10 independent trials. We perform this experiment across all search spaces and models with respect to group fairness metrics DDSP and DEOD. We don't include results on the Adult dataset as single evaluations of EGR exceeded our 24-hour time budget.

In Figure 8, we observe the relative fairness-accuracy objective space locations of MaO Pareto Fronts (black circles) compared to hyperparameter configurations post-processed with EGR (pink crosses) to minimize DDSP. Overall, we observe ManyFairHPO to be quite competitive with EGR, and in

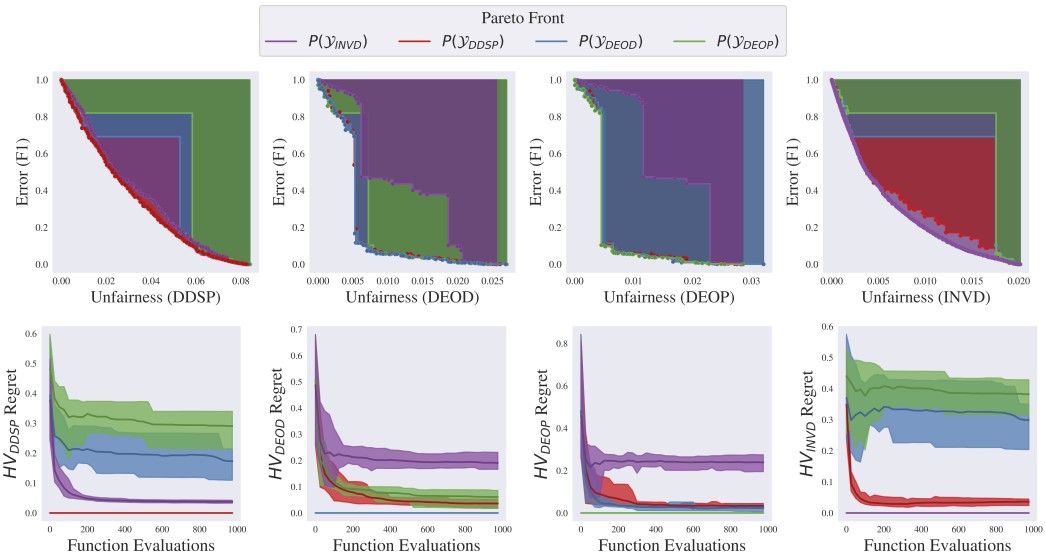

Figure 7: **Conflict Explanation (RF-Adult)**: Hypervolume regret (top) and visualization of fairness metric conflicts (bottom) observed on the RF-Adult experiments. Optimizing for a particular fairness metric does not necessarily optimize for other potentially relevant notions.

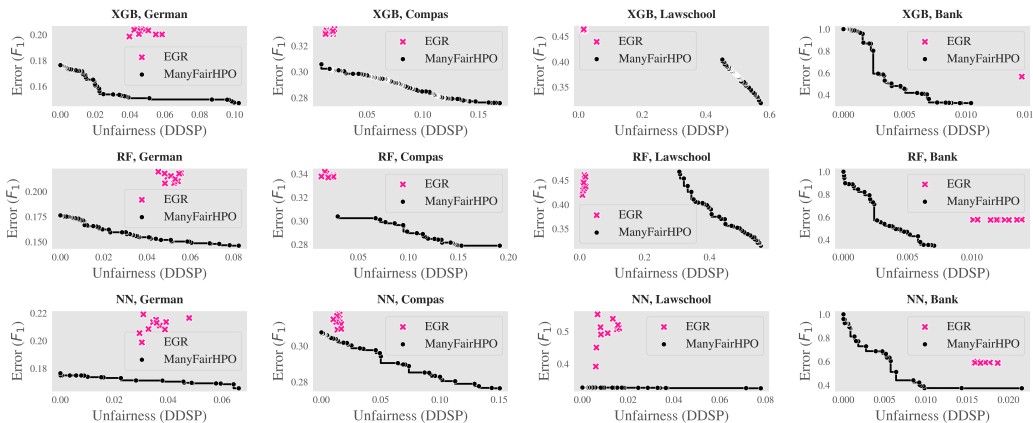

Figure 8: **Comparison to Bias Mitigation (DDSP)**: Relative fairness-accuracy objective space locations of hyperparameter configurations found by ManyFairHPO and those post-processed with Exponentiated Gradient Reduction (EGR) to minimize DDSP. ManyFairHPO Pareto Fronts dominate EGR models in the majority of cases (9/12), suggesting that HPO alone is a competitive approach to bias-mitigation.

many cases (9/12), EGR is dominated by our Pareto Fronts. We observe similar results in Figure 9, where ManyFairHPO dominates EGR (DEOD) in 7/12 cases. Although we highlight that the purpose of this study is not to specifically compare the performance of bias mitigation strategies (but rather to evaluate the socio-technical benefits of the MaO problem formulation for fairness), this outcome increases the trustworthiness of our experimental results 6.

## A.4 ASYMMETRY OF FAIRNESS METRIC CONFLICTS

In this section, we outline a scenario where a substantial difference in base rates leads to an asymmetric fairness metric conflict. An asymmetric fairness metric conflict occurs when the impact of

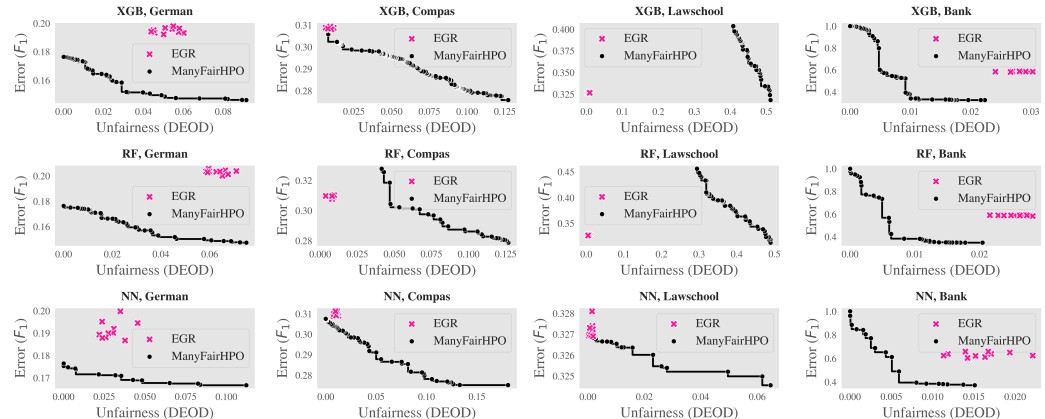

Figure 9: **Comparison to Bias Mitigation (DEOD)**: Relative fairness-accuracy objective space locations of hyperparameter configurations found by ManyFairHPO and those post-processed with Exponentiated Gradient Reduction (EGR) to minimize DEOD. ManyFairHPO Pareto Fronts dominate EGR models in the majority of cases (7/12), suggesting that HPO alone is a competitive approach to bias-mitigation.

satisfying fairness metric $f_i$ on violating fairness metric $f_j$ is different (larger or smaller) from the impact of satisfying $f_j$ on violating $f_i$.

In Figure 2, we observe an interesting phenomenon on the RF-Lawschool experiment, where the conflict between INVD with respect to group fairness metrics DDSP and DEOP/D ($C \approx 0.2$) is significantly larger than the conflict between group fairness metrics DDSP and DEOP/D with respect to INVD ($C \approx 0.1$). In plainer terms, if INVD is satisfied it leads to a strong violation of DDSP, while the satisfaction of group fairness metrics leads to a relatively weaker violation of INVD.

In the following argument, we explain why asymmetry occurs on the Lawschool dataset by drawing a connection to the significant imbalance (92% White and 8% Black) in privileged and unprivileged applicants (Appendix Table 4). Consider a perfect classifier that satisfies INVD by accepting all qualified applicants and rejecting all unqualified ones. Referring to the distribution in Appendix Figure 6 (top-right), the classifier thus accepts all 1% of applicants who are qualified and Black as well as all 50% who are qualified and White. Similarly, the classifier rejects all 7% of applicants who are unqualified and Black as well as all 42% who are unqualified and White. Although INVD is satisfied (all individuals receive the outcome they deserve, regardless of demographic group), such an admissions strategy strongly violates DDSP (precisely resulting in $P(Accept|White) - P(Accept|Black) = \frac{42}{92} - \frac{1}{8} = 0.34$ or a difference in between-class acceptance rate of 34%).

Now consider modifying this classifier (e.g. with a postprocessing technique) such that DDSP is satisfied by increasing the acceptance likelihood for unqualified Black students by 3% and decreasing the acceptance likelihood for qualified White students by 4% (positive discrimination), resulting in $P(Accept|White) - P(Accept|Black) = \frac{46}{92} - \frac{4}{8} = 0$. Such a modified classifier results in only a 24% violation of INVD, as 3% and 4% of similarly qualified (or unqualified) applicants from different demographic groups receive different admissions/rejection outcomes.

Note that the impact of DDSP on INVD depends on the base rate of privileged/unprivileged applicants, and asymmetry would increase in this scenario if the *overall* proportion of Black applicants increased while the ratio of qualified and unqualified Black applicants stayed the same. For example, if 2% of applicants were qualified and Black, while 14% of applicants were unqualified and Black, satisfying DDSP, would require a 6% (as opposed to the previous 3%) increase in acceptance likelihood for unqualified Black applicants, leading to a larger increase in INVD than in the previous example. We thus exemplify how fairness metric conflicts can be asymmetric, while also identifying the impact that dataset characteristics (e.g. difference in base rates) can have on their occurrence, significance, and symmetry. This identification suggests that fairness metric conflicts can potentially be anticipated during domain-knowledge-driven deliberations, adding a technical and concrete angle to these discussions.

# B  EXPERIMENTAL DETAILS

## B.1  MULTI-CRITERIA OBJECTIVE FUNCTION

Our objective function takes as input a hyperparameter configuration $\lambda \in \Lambda$, a FairML data set $\mathcal{D} = (X, Y, A)$, and a subset of the fairness metrics $\{f_0, f_1, f_2, ..., f_d\}$. The objective function applies Nested Stratified k-Fold Cross-Validation to iteratively partition the data set into training, testing, and validation folds $\mathcal{D}_{train}, \mathcal{D}_{val}$ and, $\mathcal{D}_{test}$. Each fold is stratified by both the target $Y$ and protected attribute $A$ in order to maintain a realistic distribution of these variables.

Given a candidate hyperparameter configuration $\lambda \in \Lambda$, a model $\mathcal{M}$ is defined and fit to the training fold $\mathcal{D}_{train}$, generating predictions $\hat{Y}$ on the validation set $\mathcal{D}_{val}$. The predictive performance of the hyperparameter configuration $f_0(Y, \hat{Y})$ is calculated using the $F_1$-Score, an appealing performance metric in the face of significant class imbalance. Because a higher $F_1$-Score is better with respect to predictive performance and defined in the range $(0, 1)$, we minimize $f_0(Y, \hat{Y}) := 1 - F_1$ during optimization. The *unfairness* of the hyperparameter configuration $f_{1:d}(Y, \hat{Y}, A)$ is calculated using the measures of fairness defined in Table 2. The objective values of each evaluated hyperparameter configuration are added to an archive of all observations $\mathcal{Y}$.

## B.2  BI-OBJECTIVE OPTIMIZATION

Motivated by the well-known theoretical conflicts between different notions of fairness, we leverage our bi-objective experiments in order to quantify the *contrast* between fairness objectives, or the degree to which optimizing for a particular fairness metric optimizes for another.

For each fairness metric $f_i$, $1 \leq i \leq d$, the archive $\mathcal{Y}_i$ is obtained by the associated bi-objective optimization run. Next, we compute the Pareto Front $\mathcal{P}(\mathcal{Y}_i)$ with respect to the $F_1$-Score and the fairness metric $f_i$.[4] To assess the performance of the Pareto Front with respect to a different fairness metric $f_j$, we compute the normalized hypervolume $\mathcal{H}_{f_j}(\mathcal{P}(\mathcal{Y}_i))$ with respect to the $F_1$-Score and the other fairness metric $f_j$. This allows us to introduce the following notion of fairness metric *contrast*.

**Definition B.1** (Contrast). The *contrast* of fairness metric $f_i$ with respect to fairness metric $f_j$ is defined as the difference in normalized hypervolume when optimizing for $f_j$ and $f_i$, respectively:

$$C(f_i, f_j) := \mathcal{H}_{f_j}\big(\mathcal{P}(\mathcal{Y}_j)\big) - \mathcal{H}_{f_j}\big(\mathcal{P}(\mathcal{Y}_i)\big) \tag{4}$$

Note that the contrast metric is not symmetric, and we provide an example of a scenario where this is the case in Appendix Section **??**. A large (positive) value of $C(f_i, f_j)$ indicates a severe conflict between notions of fairness, where optimizing for fairness metric $f_i$ fails to optimize for fairness metric $f_j$. A contrast value close to zero indicates a weak conflict and a negative contrast value indicates a *negative* conflict, where optimizing for another notion of fairness is better than optimizing for one directly.

## B.3  EVOLUTIONARY ALGORITHMS

Many design spaces are large and non-differentiable, rendering an exhaustive search or gradient-based methods computationally intractable. A popular approach to MOO problems is the Evolutionary Algorithm (EA), a population-based optimization technique that draws inspiration from the process of biological evolution to solve black-box, non-differentiable optimization problems (Eiben and Smith, 2015). By implementing bio-evolutionary concepts such as selection, mutation, and crossover, EAs effectively balance exploration and exploitation, generating state-of-the-art results in a variety of domains.

Nondomoninated-Sorting Genetic Algorithm (NSGA-II) is a state-of-the-art MO-EA (Deb et al., 2002), which applies the notion of dominance in order to recursively divide the population into

---

[4]Note that the Pareto Front $\mathcal{P}(\mathcal{Y}_i)$ remains a function in all $d$ fairness metrics together with $F_1$ and thus it is the Pareto Front with respect to objectives $F_1$ and $f_i$ and not necessarily with respect to $F_1$ and $f_j$.

ranked fronts. Parent and survivor selection is performed using the heuristics of front rank and crowding distance, which measures the distance of solutions to their nearest neighbor on their respective front. Entire fronts are greedily selected until including the next front exceeds a predefined quota. At this stage, individuals are selected from the final front based on crowding distance, which serves as a heuristic for uniqueness, and encourages exploration through population diversity. NSGA-III (Deb and Jain, 2013) incorporates the concept of reference directions in order to select individuals for crossover and survival from equally spaced regions of the many-dimensional objective space.

For our MO and MaO experiments, we opt for the `pymoo` implementations of NSGA-II and NSGA-III. For NSGA-II, we set the population size to 20 and for NSGA-III we set the number of reference directions to its default of 10, and apply Das and Dennis' approach (Das and Dennis, 1998) to define well-spaced reference points, resulting in a population size of 42 individuals. In order to provide a fair comparison, we run NSGA-II and NSGA-III for the same number of function evaluations (1000) and set all other optimizer hyperparameters to their default values. A summary of our experimental details is provided in Appendix Table 3.

### B.4 Hypervolume Regret

Regret is a concept often used in ML research to quantify the performance of an algorithm when the optimal performance is unknown. Regret is then calculated by comparing the *best that we can do* with *how we did*, and offers an estimate of algorithmic performance.

In this study, we define two forms of regret, both of which assume that the *best we can do* is optimizing for $F_1$-Score and a fairness metric $f_j$ directly. MO regret ($R_{MO}$) corresponds closely to the notion of fairness metric contrast (Equation 4) at different stages $t$ of optimization, and measures the difference in $\mathcal{H}_{f_j}$ between optimizing for fairness metric $f_j$ directly and optimizing for another fairness metric $f_i$.

$$R_{MO} := \mathcal{H}_{f_j}\big(\mathcal{P}(\mathcal{Y}_j)\big) - \mathcal{H}_{f_j}\big(\mathcal{P}(\mathcal{Y}_i^t)\big) \tag{5}$$

Similarly, MaO regret ($R_{MaO}$) measures the loss in $\mathcal{H}_{f_j}$ incurred from optimizing for all fairness metrics $f_{1:d}$ simultaneously by selecting solutions at different time steps $t$ from the many-dimensional Pareto Front $\mathcal{P}(\mathcal{Y}_{1:d})$.

$$R_{MaO} := \mathcal{H}_{f_j}\big(\mathcal{P}(\mathcal{Y}_j)\big) - \mathcal{H}_{f_j}\big(\mathcal{P}(\mathcal{Y}_{1:d}^t)\big) \tag{6}$$

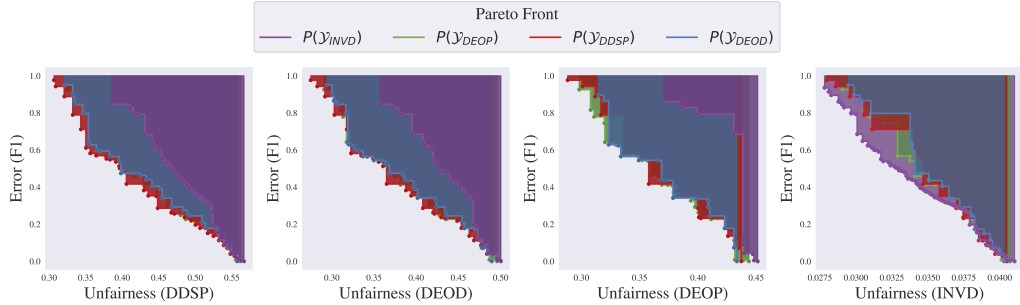

Figure 10: **Conflict Explanation (RF-Lawschool)**: Visualization of fairness metric conflicts observed on the RF-Lawschool experiment. Optimizing for $F_1$-Score and INVD does not optimize for $F_1$-Score and measures of group fairness, especially towards low unfairness regions of the objective space.

**Random Forest (NN)**

| Name | Range | Scale |
|------|-------|-------|
| max_depth | (1, 50) | Log |
| min_samples_fold | (2, 128) | Log |
| min_samples_leaf | (1, 20) | Uniform |
| max_features | (0, 1) | Uniform |
| n_estimators | (1, 200) | Log |

**XGBoost (XGB)**

| Name | Range | Scale |
|------|-------|-------|
| eta | $(2^{-10}, 1.0)$ | Log |
| max_depth | (1, 50) | Log |
| colsample_bytree | (0.1, 1.0) | Uniform |
| reg_lambda | $(2^{-10}, 2^{10})$ | Log |
| n_estimators | (1, 200) | Log |

**Multi-Layer Perceptron (NN)**

| Name | Range | Scale |
|------|-------|-------|
| depth | (1, 3) | Uniform |
| width | (16, 1024) | Log |
| batch_size | (4, 256) | Log |
| alpha | $(10^{-8}, 1)$ | Log |
| learning_rate_init | $(10^{-5}, 1)$ | Log |
| n_iter_no_change | (1, 20) | Log |

Table 1: **HPO Search Spaces**: Summary of hyperparameter search spaces drawn from HPOBench.

| Name | Definition |
|------|-----------|
| Demographic Statistical Parity (DDSP) | $\|P(\hat{Y} = 1\|A = 0) - P(\hat{Y} = 1\|A = 1)\|$ |
| Equalized Opportunity (DEOP) | $\|P(\hat{Y} = 1\|A, Y = 0, 1) - P(\hat{Y} = 1\|A, Y = 1, 1)\|$ |
| Equalized (Average) Odds (DEOD) | $\frac{1}{2} \sum_{y \in \{0,1\}} \cdot \|P(\hat{Y} = 1\|A, Y = 0, y) - P(\hat{Y} = 1\|A, Y = 1, y)\|$ |
| Inverse Distance (INVD) | $\frac{1}{m^2} \cdot \sum_{i,j=1}^{m} \|y_i - y_j\| \cdot \|\hat{y}_i - \hat{y}_j\|$ |

Table 2: **Fairness metrics**: Summary of unfairness measures drawn from the aif360 library. Inverse Distance (INVD) is a simplified version of similarity-based individual fairness that does not require the definition of problem-specific inverse-distance functions.

| Formulation | Name | Optimizer | Objectives | Pop. Size | Func. Evals. | Seeds |
|-------------|------|-----------|-----------|-----------|--------------|-------|
| MO | F1-DDSP | NSGA-II | 2 | 20 | 1000 | 10 |
| | F1-DEOD | NSGA-II | 2 | 20 | 1000 | 10 |
| | F1-DEOP | NSGA-II | 2 | 20 | 1000 | 10 |
| | F1-INVD | NSGA-II | 2 | 20 | 1000 | 10 |
| MaO | F1-MULTI | NSGA-III | 5 | 42 | 1000 | 10 |

Table 3: **ManyFairHPO Experiments:** Summary of ManyFairHPO experiments, spaning across two problem formulations, four fairness metrics, three HPO search spaces, and five data sets. We run each experiment for 10 seeds with a maximum wall-clock time of 1 CPU day.

| Name | Prot. Attr. | Samples | Features | Pos./Neg. | Priv./Unpriv. |
|---|---|---|---|---|---|
| German Credit | sex | 1,000 | 59 | 70/30 | 69/31 |
| Criminal Recidvism | race | 5278 | 7 | 53/47 | 40/60 |
| Bank Marketing | age | 764 | 31 | 23/77 | 64/36 |
| Census Income | sex | 15,315 | 44 | 25/75 | 85/14 |
| Lawschool Admissions | race | 22,342 | 3 | 25/75 | 92/8 |

Table 4: **FairML Datasets:** Summary of data sets drawn from the `aif360` library.

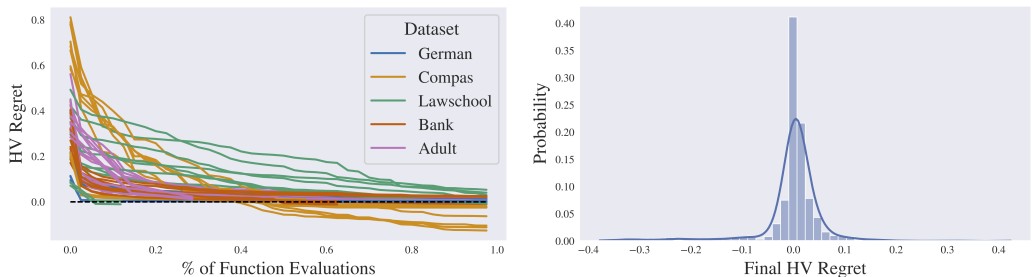

Figure 11: **Hypervolume Regret (MaO)**: Difference in normalized hypervolume between optimizing for fairness metrics together and optimizing for them seperately. ManyFairHPO is capable of efficiently exploring multiple fairness-accuracy trade-offs and typically converges to zero regret.

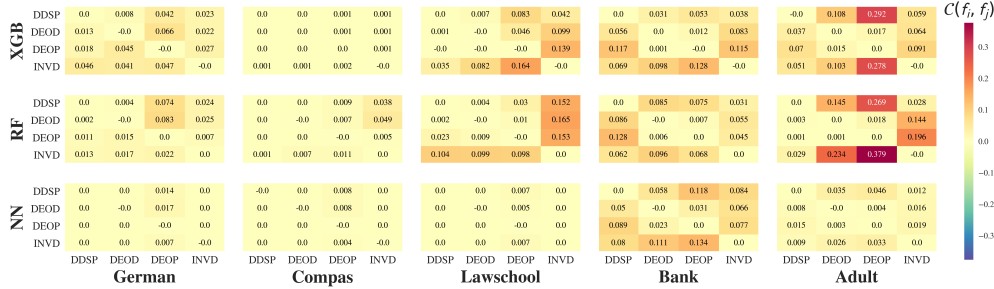

Figure 12: **Fairness Metric Conflicts (MaO)**: Overview of fairness metric conflicts calculated from MaO experiments. Fairness metric conflicts are similar between MO and MaO problem formulations but MaO does not create *negative* conflicts.

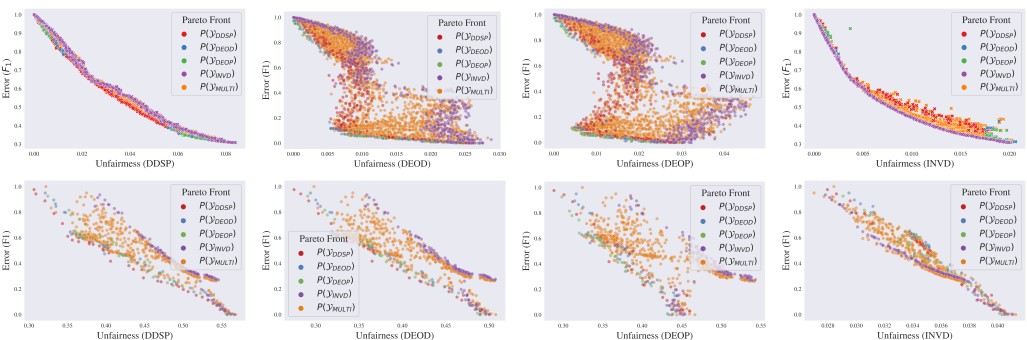

Figure 13: **MaO Pareto Fronts**: Visualization of the MaO Pareto Front in the presence of a fairness metric conflicts observed on the RF-Adult (top) and RF-Lawschool (bottom) experiments. ManyFairHPO has the consistent effect of filling the gaps left by bi-objective optimization.