# OpenReview forum: "Fairness Metric Impossibility: Investigating and Addressing Conflicts"
_ICLR.cc/2024/Conference — Submitted to ICLR 2024_

### Official Review · Reviewer_RKvV · 2023-10-31

**Soundness:** 3 good
**Presentation:** 4 excellent
**Contribution:** 2 fair
**Rating:** 6
**Confidence:** 5

**Summary:**

In this manuscript, the authors propose and present a many-objective optimization approach to fair ML, where models are optimized simultaneously for a standard performance metric (F1 score) as well as multiple fairness-motivated metrics. Standard tools from many-objective optimization are used (NSGA-III, an evolutionary algorithm) to solve the optimization problem, and the multi-dimensional fairness-accuracy Pareto frontiers resulting from the specified hyperparameter search space are analyzed.

Throughout the manuscript, particular emphasis is placed on incompatibilities between different fairness metric: are two metrics indeed conflicting, and if yes, how strongly? The authors propose a way to quantify the strength of the association between two metrics in terms of Pareto set hypervolume contrasts.

The authors apply their method to three different model structures - XGBoost, Random Forests, and shallow MLPs - on five standard (tabular) algorithmic fairness datasets. They show a range of visualizations of the Pareto frontiers as well as the conflicts (or compatibilities) between all optimization objectives. They conclude with a call for multi-objective optimization-based exploration of potentially conflicting fairness objectives as a standard analysis step in fair ML.

**Strengths:**

The paper is very well written and easy to follow.

The idea is simple, intuitive, and widely applicable, and the visualizations are very helpful in understanding the extent of practical conflicts between different fairness objectives.

The case studies, while limited to tabular data and rather low-dimensional models, showcase the broad applicability of the proposed method.

Standard methodology is used throughout, and I would expect the results to be relatively simple to reproduce.

**Weaknesses:**

## Many-objective optimization as the solution to fairness metric conflicts
I am generally very receptive to the idea of using MOO in fairML, both for interactive exploration and visualization of trade-offs, as well as for training and selecting the final model. I am, however, concerned about the broader framing adopted in the paper.

The manuscript begins by acknowledging the well-known fairness impossibility theorems. The authors then proceed to make statements such as:
- "We open the door to further socio-technical research on effectively combining the complementary benefits of different notions of fairness"
- "Rather than resisting these criticisms, we embrace them, agreeing that FairML approaches that over-simplify the complex and socio-technical nature of the FairML problem actually risk doing more social harm than good."
- "It is essential to recognize and treat fairness metrics as potentially conflicting objectives."

The authors then proceed to take a more or less arbitrary selection of fairness metrics and apply them indiscriminately to five different datasets.

However, I would contest that this is the right approach to dealing with these impossibility statements (of which there are less than is often believed*). There is a reason why, for instance, demographic parity and equalized odds are fundamentally incompatible: *they are asking for completely different things*. They are also applicable (=usually considered ethically desirable) in completely different scenarios. There are no "complementary benefits" of enforcing these simultaneously; I cannot think of a real situation where one would want a little bit of both. Enforcing statistical parity in a medical scenario will give you a classifier that predicts breast cancer on 50% of men and 50% of women, even though women have a much higher disease incidence, and provide no benefit at all.

The way to address incompatibility results is, in my opinion, not to mash all of them together and hope that the resulting model will be somewhat "fair" according to all metrics; it is to understand and reflect on the *reasons* why certain metrics are incompatible, and to deliberate with stakeholders and practitioners which fairness conception would appear to be the right one for a given application scenario.

With all that said, I still believe that MOO can be a very useful tool! I can certainly imagine it being very useful for interactive explorations of various trade-offs between multiple metrics that have been selected as indeed desirable in a certain application. I would, however, suggest that the authors place more emphasis on *understanding* the different fairness metrics, and actively deciding on the appropriate ones for a given application, before embarking on fully-automated many-objective optimization of all of them simultaneously.

In this regard, I would also suggest choosing a more realistic combination of fairness metrics for the application scenarios, or at least a cursory justification of the ones selected here. (Also note in this regard that DEOP is a subset of DEOD, and selecting both of them as simultaneous but separate optimization objective appears a bit odd.)

Many of the experimental results might also make more sense if interpreted based on knowledge about what these metrics mean. For instance, the harms caused by enforcing statistical parity depend on the magnitude of the prevalence differences between the different groups in the dataset (Zhao and Gordon, 2022). This might explain some of the differences observed between the five datasets, such as in Fig. 2. (The group-stratified prevalences are currently not given in the manuscript, so this is hard to assess right now.) Similarly, it might be expected that DDSP and INVD do not seem to be in conflict, since they probably optimize for something very similar. (I could not figure out what exactly INVD does - what is "m" in table 2 in the appendix, and what are the sets being summed over?)

*Equalized odds and calibration by groups and AUROC fairness are all compatible in principle, see e.g. Lazar Reich and Vijaykumar (2020) or Petersen et al. (2023). They are all at odds with statistical parity, which simply does not make any sense in any predictive scenario and is fully incompatible with any predictive performance-based fairness notion (Zhao and Gordon, 2022). They are also in conflict with PPV/NPV equality, which simply does not make sense to ask for in the case of prevalence differences. For a predictive performance fairness setting, I am not aware of any meaningful fundamental impossibility statements between different actually desirable fairness metrics.

## Prior work and significance of contributions
Fairness-related trade-offs and Pareto frontiers have already received significant attention in the literature; cf., e.g., Cooper et al. (2021), Islam et al. (2021), Martinez et al. (2020),  Rodolfa et al. (2021), Wei and Niethammer (2021), Yu et al. (2020).

What does the present study contribute to this already quite large body of literature? I would venture to say: a useful, practical tool, as well as appropriate metrics, for exploring such trade-offs in a given dataset. That is most certainly a useful contribution (even though probably limited to rather small and simple cases with low-dimensional models?), but then I would recommend discussing this prior work more extensively, and clarifying the contributions of the present manuscript in this regard.

Finally, I am not entirely certain about the topical fit of this piece for ICLR, seeing that the manuscript is not at all focused on representation learning. (Fairness constraints will, of course, affect the learned representations. However, these learned representations are also not assessed in any way in the study, and, again, only low-dimensional, tabular case studies are considered.)

## References
- Cooper et al. (2021), Emergent Unfairness in Algorithmic Fairness-Accuracy Trade-Off Research, https://dl.acm.org/doi/abs/10.1145/3461702.3462519
- Islam et al. (2021), Can we obtain fairness for free?, https://dl.acm.org/doi/abs/10.1145/3461702.3462614
- Lazar Reich and Vijaykumar (2020), A Possibility in Algorithmic Fairness: Can Calibration and Equal Error Rates Be Reconciled?, https://drops.dagstuhl.de/opus/volltexte/2021/13872
- Martinez et al. (2020), Minimax Pareto Fairness: A Multi Objective Perspective, http://proceedings.mlr.press/v119/martinez20a.html
- Petersen et al. (2023), On (assessing) the fairness of risk score models, https://arxiv.org/pdf/2302.08851.pdf
- Rodolfa et al. (2021), Empirical observation of negligible fairness–accuracy trade-offs in machine learning for public policy, https://www.nature.com/articles/s42256-021-00396-x
- Wei and Niethammer (2021), The fairness-accuracy Pareto front https://onlinelibrary.wiley.com/doi/full/10.1002/sam.11560
- Yu et al. (2020), Keeping Designers in the Loop: Communicating Inherent Algorithmic Trade-offs Across Multiple Objectives, https://doi.org/10.1145/3357236.3395528
- Zhao and Gordon (2022), Inherent Tradeoffs in Learning Fair Representations, https://jmlr.org/papers/volume23/21-1427/21-1427.pdf

**Questions:**

--

---

> ### Author Response · Authors · 2023-11-14
>
> > I am generally very receptive to the idea of using MOO in fairML, both for interactive exploration and visualization of trade-offs, as well as for training and selecting the final model. I am, however, concerned about the broader framing adopted in the paper.
>
> We thank you for this incredibly valuable feedback regarding the groundedness of our work with respect to the fairness incompatibility theorem. We would like to point out that ManyFairHPO is a general framework that should be specifically applied to different fair machine learning scenarios. In other words, we do not propose that practitioners optimize for an indiscriminate set of fairness metrics, and choose a model that is relatively “fair” in terms of all of them. On the contrary, we believe that the involvement of domain experts in the discussion of which fairness metrics to optimize for is crucial, and should also be accompanied with discussion of what the risks are if one notion of fairness is satisfied by violating another relevant notion. With that being said, we acknowledge that we should be clearer about how ManyFairHPO should be used in practice, and, we have updated our methodology to include a more careful deliberation on how ManyFairHPO should be applied in practice (see Figure 1 and Section 4). We have also provided an exemplary use-case to show the efficacy of our approach in providing solutions in the face of many conflicting fairness objectives and conflict related risks (Section 6.3). We believe that our modified and extended approach clarifies that each FairML problem is categorized by a unique set of objectives and risks, and the MaO problem formulation we propose is extremely useful in such scenarios. To be clear, we do not seek to automate the FairML process, but provide a socio-technical approach that allows practitioners to make informed model selection in FairML problems that are categorized by social objectives that span across multiple conflicting notions of fairness..

---

> ### Author Response · Authors · 2023-11-14
>
> > Fairness-related trade-offs and Pareto frontiers have already received significant attention in the literature; cf., e.g., Cooper et al. (2021), Islam et al. (2021), Martinez et al. (2020), Rodolfa et al. (2021), Wei and Niethammer (2021), Yu et al. (2020).
> What does the present study contribute to this already quite large body of literature? I would venture to say: a useful, practical tool, as well as appropriate metrics, for exploring such trade-offs in a given dataset. That is most certainly a useful contribution (even though probably limited to rather small and simple cases with low-dimensional models?), but then I would recommend discussing this prior work more extensively, and clarifying the contributions of the present manuscript in this regard.
>
> Yes, as you said! We appreciate your suggestion to clarify the contribution of our work. We will include your suggestions into our related work section. As we have emphasized in our general reply, we are the first work that treats fairness metrics as conflicting objectives, and thoroughly evaluates this methodology.

---

> ### Comment · Reviewer_RKvV · 2023-11-22
> **Response to rebuttal**
>
> I would like to thank the authors for their quite substantial rebuttal and revision of the paper!
>
> I really like the new section 4.0.1 + Fig.1 / the revised framing in terms of domain knowledge-driven deliberation for picking fairness metrics; I feel like this works a lot better now and provides a much more thorough and grounded discussion.
>
> **As a result, I am increasing my score.** The paper is well written, and the results, experiments and analyses are quite comprehensive. I do think that this adds something useful to the literature, even though a few questions currently remain open (see below).
>
> A few more things that could still be improved:
> 1) There is still no discussion of _why_ specific fairness criteria might be incompatible. In particular, I still believe that a (short) discussion of different base rate differences across the different datasets as a potential explanation of some of the observed fairness conflicts w.r.t. DDSP would add an essential piece of the puzzle.
> 2) In regards to table 2, there are two more or less obvious questions: i) Why are the conflicts so much stronger in some models compared to others, and ii) why are the conflict matrices so asymmetric, e.g. in the COMPAS/RF, Adult/RF, Adult/XGB cases? It would also be very helpful to add information about the predictive performance of the models in the table. E.g. NNs are known to under-perform on tabular data; could that somehow be related to the fact that we see fewer conflicts for the NN?
> 3) Could Fig. 4 maybe be augmented with a second panel that shows trade-offs w.r.t. performance? Currently, it looks a bit odd that the the black model is selected and not one of those in the bottom left corner that dominate this model in terms of fairness w.r.t. both shown criteria. I suspect this is due to model performance, but this is currently not visible in the graph.
> 4) To gain some more trust in the method, it would also be useful to add one or two more standard baseline methods w.r.t. a single fairness criterion (e.g. the standard postprocessing method of Hardt et al., Equality of opportunity in supervised learning, and maybe also the equalized odds version of EGR), in order to demonstrate that the MOO framework indeed recovers optimal Pareto bounds in most cases.

---

> > ### Author Response · Authors · 2023-11-23
> >
> > > A few more things that could still be improved:
> >
> > > 1. There is still no discussion of why specific fairness criteria might be incompatible. In particular, I still believe that a (short) discussion of different base rate differences across the different datasets as a potential explanation of some of the observed fairness conflicts w.r.t. DDSP would add an essential piece of the puzzle.
> > > 2. In regards to table 2, there are two more or less obvious questions: i) Why are the conflicts so much stronger in some models compared to others, and ii) why are the conflict matrices so asymmetric, e.g. in the COMPAS/RF, Adult/RF, Adult/XGB cases? It would also be very helpful to add information about the predictive performance of the models in the table. E.g. NNs are known to under-perform on tabular data; could that somehow be related to the fact that we see fewer conflicts for the NN?
> > > 3. Could Fig. 4 maybe be augmented with a second panel that shows trade-offs w.r.t. performance? Currently, it looks a bit odd that the the black model is selected and not one of those in the bottom left corner that dominate this model in terms of fairness w.r.t. both shown criteria. I suspect this is due to model performance, but this is currently not visible in the graph.
> > > 4. To gain some more trust in the method, it would also be useful to add one or two more standard baseline methods w.r.t. a single fairness criterion (e.g. the standard postprocessing method of Hardt et al., Equality of opportunity in supervised learning, and maybe also the equalized odds version of EGR), in order to demonstrate that the MOO framework indeed recovers optimal Pareto bounds in most cases.
> >
> > We would like to sincerely thank you for your continued engagement and thoughtful and actionable suggestions! We have incorporated your open questions (new changes highlighted in yellow) as detailed below and updated the main text/appendix as follows:
> >
> > 1. You make a strong point that a more theoretical discussion regarding why fairness metric conflicts occur would improve the comprehensiveness of our overall argument, motivating the complexity of the fairness metric selection problem but also exemplifying how certain dataset characteristics (e.g.. differences in base rates) can help to anticipate certain conflicts. As such, we have incorporated a discussion (Appendix A.4) of the effect of base rates in privileged and unprivileged groups in the context of the Lawschool dataset, as a means to anticipate and explain the conflict between DDSP and INVD. We believe this toy example exemplifies the effect of dataset characteristics on fairness metric conflicts, demystifying the data-dependence of our results, and could potentially lead to future discussions regarding how to use dataset characteristics to anticipate conflict-related risks.
> > 2.
> > - In response to your inquiry regarding the effect of model performance on fairness metric conflict strength, we have annotated Figure 2 with the minimum error achieved by each model on each dataset. However, we have not found a clear signal connecting relative model performance and the strength of fairness metric conflicts discovered. Although XGB often outperforms NN and has stronger conflicts, RF sometimes has stronger conflicts (COMPAS) than XGB without increased performance.
> > - Regarding your inquiry into the symmetry (or lack thereof) of fairness metrics, we agree that this whole must be filled in our argument. We have thus included Appendix Section A.4 explaining an asymmetry we observed on the RF-Lawschool experiment. We also related this asymmetry to the significant difference in base rates of privileged/unprivileged groups (92/8), which as mentioned also responds to your first open question.
> >
> > 3. Yes we absolutely agree, the incorporation of performance into Figure 4 helps explain how fairness-accuracy trade-offs play into our model selection decision by using a color gradient on our MaO Pareto Front. Indeed, we did not select models from the bottom-left region due to poor model performance. Instead, the preference (w=0.5) towards F1-score in this exemplary use case guides us toward the top right, where models are generally more accurate, and less fair.
> > 4. We thank you for your suggestion to supplement our bias-mitigation experiments (Appendix A.3) in order to increase the trustworthiness of our experimental results. As you suggested, we have incorporated an additional experiment that applies the Equalized Odds version of Exponentiated Gradient Reduction (EGR) and compares this technique to ManyFairHPO’s Pareto Fronts. Now for both Statistical Parity and Equalized Odds metrics (Appendix Figure 7 and 8), we observe that ManyFairHPO Pareto Fronts dominate EGR 9/12 (DDSP) and 7/12 (DEOD) times. This result provides further evidence for the promise of hyperparameter optimization for fairness, and the applicability of ManyFairHPO in real-world scenarios as an alternative to single-metric/output bias-mitigation techniques.

---

### Official Review · Reviewer_k3oE · 2023-10-31

**Soundness:** 3 good
**Presentation:** 2 fair
**Contribution:** 2 fair
**Rating:** 5
**Confidence:** 4

**Summary:**

This paper studies the trade-offs among different fairness objectives in fair machine learning. Motivated by the well-known impossibility results among fairness metrics in fair ML, the authors adopted a many-objective optimization (MaO) perspective to formulate the problem of ML with multiple fairness objectives. Their main result included a ManyFairHPO framework that enables model designers to specify multiple fairness objectives, and utilizes hyperparamter optimization to select a ML model attaining desirable trade-offs among the specified objectives. On several datasets, the authors first provided new empirical evidence for the necessary trade-offs among fairness metrics in ML, then applied the ManyFairHPO framework to simultaneously consider multiple fairness metrics in the ML model. They argued that their framework is effective at reaching a superior fairness balance compared to the conventional approach of ML with a single fairness metric.

**Strengths:**

The paper provided a systematic approach for understanding and mitigating the potential conflicts among multiple fairness objectives. The ‘Contrast’ measure gives a convenient way to visualize the differences between the fairness-accuracy trade-offs under different fairness requirement. The experiment considered a broad set of datasets and ML models. In addition, the proposed framework is compatible with existing ML tool (scikit-learn) and fair ML library (IBM aif360).

**Weaknesses:**

The problem of incorporating multiple fairness objectives is well recognized in the fair ML community, and the adopted method based on multi-objective optimization is also conventional. When one wishes to consider multiple fairness goals, it is a natural attempt to apply multi-objective optimization methods to include all of them. Although there is certainly value in working on the technical details and running experiments to validate performance, I see limited novelty and depth in the proposed ManyFairHPO framework special and useful.

Another concern is that using multiple fairness objectives makes the task of model interpretation and selection more difficult. The proposed framework can give more information about whether two fairness objectives are conflicting or not, but it is not as helpful for handling the more fundamental task of what combination of the candidate fairness objectives should be used. Since there are always trade-offs among different fairness goals, instead of trying to optimize all of them, one may wish to examine the more important goals and be more selective with which fairness definitions to include. The paper may benefit from highlighting the need for caution when applying the new framework.

**Questions:**

1.	The experiments only considered the fair ML methods available from IBM aif360 library. To consider other fairness definitions or fair ML methods, how to achieve that with the proposed framework?

2.	What is the use of hyperparameter optimization in the ManyFairHPO framework? Are conventional approaches to hyperparameter tuning sufficient, or are special techniques requires to handle the new multi-objective setup?

3.	Are there practical evidence that decision makers will prefer to simultaneously optimize multiple types of fairness, rather than separately study each fairness definition and select the better one?

---

> ### Author Response · Authors · 2023-11-14
>
> > The problem of incorporating multiple fairness objectives is well recognized in the fair ML community, and the adopted method based on multi-objective optimization is also conventional. When one wishes to consider multiple fairness goals, it is a natural attempt to apply multi-objective optimization methods to include all of them. Although there is certainly value in working on the technical details and running experiments to validate performance, I see limited novelty and depth in the proposed ManyFairHPO framework special and useful.
>
> Despite the natural application of multi-objective optimization, we point out that optimizing for multiple notions of fairness as conflicting objectives has not been thoroughly evaluated in the literature. The closest case of this is [1], who argue that their framework is extensible to multiple fairness metrics, but do not include any experiments. In addition to our validation of the performance of this methodology, we also are working on contextualizing it in a theoretical framework that can allow practitioners to use it in real case scenarios. This represents an additional contribution to our paper.
>
> In addition to the distinction pointed out above, we have focused on the re-framing of ManyFairHPO. In order to do so, we have reformulated our methodology (Figure 1 and Section 4) to provide a start-to-finish framework from fairness metric selection and conflict-related risk identification to Many-Objective model selection, going even deeper into this socio-technical problem than the previously mentioned work [1].
>
> [1] https://www.amazon.science/publications/multi-objective-multi-fidelity-hyperparameter-optimization-with-application-to-fairness

---

> ### Author Response · Authors · 2023-11-14
>
> > Another concern is that using multiple fairness objectives makes the task of model interpretation and selection more difficult. The proposed framework can give more information about whether two fairness objectives are conflicting or not, but it is not as helpful for handling the more fundamental task of what combination of the candidate fairness objectives should be used. Since there are always trade-offs among different fairness goals, instead of trying to optimize all of them, one may wish to examine the more important goals and be more selective with which fairness definitions to include. The paper may benefit from highlighting the need for caution when applying the new framework.
>
> Thank you for your input regarding the inherent challenge of selecting fairness metrics. We would argue a middle ground, that practitioners should still be very selective on which fairness metrics to use for their given problem, but should also be open to the fact that problem-specific social objectives might span across multiple notions of fairness.
> We have emphasized the role of domain expert driven deliberation in ManyFairHPO (Figure 1) to clarify our framework does not propose to simply select all metrics and select a model that is somewhat fair in terms of all notions. We have also been more selective in our experiments, providing a case study (Section 6.3) that exemplifies how specific FairML problems have conflicting fairness objectives and conflict-related risks.

---

> ### Author Response · Authors · 2023-11-14
>
> > The experiments only considered the fair ML methods available from IBM aif360 library. To consider other fairness definitions or fair ML methods, how to achieve that with the proposed framework?
>
> Incorporating other fairness metrics into the ManyFairHPO framework is quite straightforward from a technical perspective. We decided to use IBM aif360 due to its popularity in the fairness community. However, end-users are encouraged to incorporate any fairness metrics that fit their specific FairML problem.

---

> ### Author Response · Authors · 2023-11-14
>
> > What is the use of hyperparameter optimization in the ManyFairHPO framework? Are conventional approaches to hyperparameter tuning sufficient, or are special techniques requires to handle the new multi-objective setup?
>
> Thank you for your question regarding the use of HPO in our framework. In order to go from the MO setting to the MaO setting (three or more objectives) we applied NSGA-III, an extension of the common NSGA-II algorithm that incorporates the concept of reference directions in order to encourage equal exploration of all objectives. However, we note that NSGA-III is a classic HPO method from 2014; our problem formulation thus does not require novel multi-objective optimization algorithms.

---

> ### Author Response · Authors · 2023-11-14
>
> > Are there practical evidence that decision makers will prefer to simultaneously optimize multiple types of fairness, rather than separately study each fairness definition and select the better one?
>
> We thank you for your inquiry regarding practical evidence that motivates the demand for our approach. The short answer is that practical evidence is limited due to the lack of approaches that optimize for multiple notions of fairness as conflicting objectives. However, our motivation for this methodology came from considering specific scenarios where multiple conflicting fairness notions might be of value (e.g. statistical parity vs. individual fairness in University admissions) and where conflicts between these notions might produce unwanted risks (e.g. self-fulfilling prophecy from positive discrimination). We believe that if decision makers want to mitigate these risks, they will strongly consider the prospect of optimizing for multiple notions of fairness.

---

> ### Comment · Reviewer_k3oE · 2023-11-20
> **Thank you for the detailed responses**
>
> I would like to thank the authors for the detailed response to my questions. I recognize and appreciate the authors' effort towards proposing a framework to allow multiple fairness perspectives to be considered simultaneously, but I am unconvinced of its value as yet another framework to including fairness requirement into ML. What might strengthen the paper's position is to apply the framework on a realistic decision problem and gather relevant stakeholders' feedback on whether and how this ManyFairHPO framework is superior to more convenient alternatives.

---

> > ### Author Response · Authors · 2023-11-20
> >
> > > I would like to thank the authors for the detailed response to my questions. I recognize and appreciate the authors' effort towards proposing a framework to allow multiple fairness perspectives to be considered simultaneously, but I am unconvinced of its value as yet another framework to including fairness requirement into ML. What might strengthen the paper's position is to apply the framework on a realistic decision problem and gather relevant stakeholders' feedback on whether and how this ManyFairHPO framework is superior to more convenient alternatives.
> >
> > We thank you for your suggestion to consider the application in a real-world FairML problem, and we’ve been working on it! In addition to updating our methodology to clarify how ManyFairHPO should be applied in the real world (Section 4), we have put this into practice (Section 6.3) to exemplify the efficacy of ManyFairHPO in capturing a complex set of fairness objectives and downstream risks. We look forward to your thoughts on our updated methodology and experiments!
> >
> > Unfortunately, it is out of the scope of this rebuttal to gather stakeholder data on the benefits of our approach. This is rather our motivation to publish this work, to present something new and potentially useful to the fairness community and hear their response. We note that ManyFairHPO is not as convenient as other (single metric) bias mitigation approaches, but neither is the complex socio-technical problem of algorithmic bias. We propose an approach that, we believe, better captures the complexity of real-world FairML problems, enabling practitioners to think about fairness as a complex, Many-Objective problem.

---

### Official Review · Reviewer_XwAL · 2023-11-01

**Soundness:** 2 fair
**Presentation:** 2 fair
**Contribution:** 2 fair
**Rating:** 3
**Confidence:** 4

**Summary:**

In this paper, the authors discuss challenges in Fairness-aware Machine Learning (FairML), highlighting that optimizing for a single fairness objective can lead to neglect of other important fairness criteria. To address this, the authors introduce ManyFairHPO, a new many-objective hyper-parameter optimization framework, aimed at balancing multiple fairness objectives and exploring various fairness-accuracy trade-offs. The results on five real-world datasets are present.

**Strengths:**

1.	The paper addresses the complexities of machine learning fairness by considering multiple, potentially conflicting notions of fairness.

2.	 The authors have conducted extensive evaluations of their proposed framework across multiple real-world datasets. This comprehensive testing not only demonstrates the framework's versatility and adaptability to different contexts and applications but also adds credibility and robustness to the presented results.

3.	The incorporation of visual representations effectively communicates the experimental results, making it easier for readers to comprehend the performance and benefits of the proposed method.

**Weaknesses:**

1.	While the authors propose optimizing model fairness through the Pareto frontier and simultaneous measurement of multiple fairness indicators, they do not provide a theoretical demonstration of how trading off one fairness metric for another could lead to an overall improvement in model fairness. A deeper theoretical exploration in this area could strengthen the paper, offering clearer guidelines on how to navigate fairness trade-offs effectively.

2.	The paper lacks theoretical analysis on how to select among different Pareto-optimal outcomes, especially when one fairness metric  is already at its optimal is one of the Pareto-optimal outcomes, i.e., there is no difference from a single optimization outcome. A theoretical framework or set of criteria for making these choices would be beneficial, providing practitioners with a robust method for decision-making in situations with multiple optimal fairness solutions.

3.	The authors only use one performance to evaluate the model performance. In the context of FairML, where the applications are intricate and multifaceted, relying on a single performance metric may not sufficiently capture the model’s overall performance and impact. A diverse set of performance metrics would provide a more holistic view, ensuring a balanced and thorough evaluation.

4.	In the experimental section, the authors have not conducted comparisons with existing fairness algorithms. Integrating benchmark comparisons against state-of-the-art fairness algorithms would significantly enhance the paper. It would offer tangible evidence of the proposed method's performance and effectively position the ManyFairHPO framework within the existing FairML research landscape.

**Questions:**

In the paper, the authors propose optimizing model fairness through the Pareto frontier by measuring multiple fairness indicators simultaneously. I appreciate it if the authors could provide more theoretical insights or guidelines on how trading off one fairness metric for another could lead to an overall improvement in model fairness? Specifically, how should practitioners approach situations where improving one aspect of fairness might lead to a decrease in another, and how can they ensure that these trade-offs result in a net positive impact on model fairness?

---

> ### Author Response · Authors · 2023-11-14
>
> > 1. While the authors propose optimizing model fairness through the Pareto frontier and simultaneous measurement of multiple fairness indicators, they do not provide a theoretical demonstration of how trading off one fairness metric for another could lead to an overall improvement in model fairness. A deeper theoretical exploration in this area could strengthen the paper, offering clearer guidelines on how to navigate fairness trade-offs effectively.
>
> Thank you for your insight regarding a theoretical framework on effectively navigating fairness trade-offs. We are actively working on an intended use case which should clear up how MaO model selection works in the context of multiple fairness metrics, conflicts, and conflict-related risks. Regarding your comment regarding overall fairness, we would like to provide a clarification that trading off one fairness metric for another doesn’t lead to an overall improvement in fairness, but rather aims to mitigate the risks of fairness metric incompatibility.

---

> ### Author Response · Authors · 2023-11-14
>
> >  2. The paper lacks theoretical analysis on how to select among different Pareto-optimal outcomes, especially when one fairness metric is already at its optimal is one of the Pareto-optimal outcomes, i.e., there is no difference from a single optimization outcome. A theoretical framework or set of criteria for making these choices would be beneficial, providing practitioners with a robust method for decision-making in situations with multiple optimal fairness solutions.
>
> We thank you for this extremely valuable insight, and align that our MaO scenario makes model selection especially difficult to imagine. We have elected to incorporate a clarification into our central methodology, and have added a new Many-Objective Model Selection step to ManyFairHPO (Section 4.3), which details the use of objective weight scalarization to transform differing preferences towards multiple fairness metrics into a concrete model selection decision. We also provide a concrete example of our model selection framework on the Lawschool Admissions problem, exemplifying how ManyFairHPO can be used on complex real-world FairML problems with multiple fairness objectives and risks (Section 6.3).

---

> ### Author Response · Authors · 2023-11-14
>
> > 4. In the experimental section, the authors have not conducted comparisons with existing fairness algorithms. Integrating benchmark comparisons against state-of-the-art fairness algorithms would significantly enhance the paper. It would offer tangible evidence of the proposed method's performance and effectively position the ManyFairHPO framework within the existing FairML research landscape.
>
> Thank you for your constructive feedback to include a comparison to specially designed bias-mitigation techniques. We have included a comparison of our MO Pareto Fronts to bias mitigation techniques in Appendix Figure 8, where we postprocess high-accuracy hyperparameter configurations with the SOTA in-processing method Exponentiated Gradient Reduction (EGR). We find that ManyFairHPO is competitive with EGR, finding dominating solutions in many scenarios. Although our study is not concerned with the specific performance of bias-mitigation methods (and rather the overarching problem formulation best suited to FairML problems), we believe that this result validates the performance of ManyFairHPO and motivates the importance of HPO in fairness-aware ML applications.

---

> ### Author Response · Authors · 2023-11-20
>
> > The authors only use one performance to evaluate the model performance. In the context of FairML, where the applications are intricate and multifaceted, relying on a single performance metric may not sufficiently capture the model’s overall performance and impact. A diverse set of performance metrics would provide a more holistic view, ensuring a balanced and thorough evaluation.
>
> We thank you again for your suggestion. We find it crucial (as in any ML problem) to select appropriate performance metrics that effectively capture the utility of the algorithm in its specific application domain. While we emphasize in newly updated Section 4.1 that performance metric selection is a critical decision (and many different performance metrics can be applied), we maintain that performance-metric trade-offs are not the central focus of our paper, and open the door for future study in this direction. There is also a clear connection between fairness metrics and different aspects of performance (as many fairness metrics can be derived from false positives rate (FPR), false negatives rate (FNR), etc.), and we suggest instead that the fairness-specific impact of models be determined by carefully selecting fairness metrics and thoroughly considering their trade-offs. Having said that, our improved approach now can be viewed as a general one in which many objectives (performance and fairness metrics) can be used with careful consideration by the domain experts.

---

### Official Review · Reviewer_bcZi · 2023-11-03

**Soundness:** 3 good
**Presentation:** 3 good
**Contribution:** 2 fair
**Rating:** 5
**Confidence:** 3

**Summary:**

The paper proposed ManyFairHPO framework, a many-objective (MaO) hyper-parameter optimization (HPO) approach to study and balance the tradeoff between different fairness metrics.

**Strengths:**

1. The paper is well-written and organized.
2. The paper is well-motivated.
3. Experiments are comprehensive.

**Weaknesses:**

1. The biggest problem of the paper is the lack of novelty and technical contributions. The proposed ManyFairHPO does not show any technical improvement *adapted to fairness* compared to the prior work on multi-objective optimization or hyperparameter optimization.
2. The authors mentioned constrained optimization (CO) approaches in the related work. However, such a baseline is missing in the empirical evaluation. It would be better to compare CO with MaO empirically and provide more insights into the pros and cons of both approaches.

**Questions:**

N/A

---

> ### Author Response · Authors · 2023-11-14
>
> > 1. The biggest problem of the paper is the lack of novelty and technical contributions. The proposed ManyFairHPO does not show any technical improvement adapted to fairness compared to the prior work on multi-objective optimization or hyperparameter optimization.
>
> Thank you for your constructive criticism regarding the contribution of this work. We have addressed this matter in our 'general response' statement; kindly refer to it for details. We made significant changes to our methodology section, which we detail as follows: We have focused our efforts on grounding our approach to real-world applications, and providing a start-to-finish framework to guide practitioners in the use of our approach. ManyFairHPO now contains guidelines on how to select and prioritize fairness metrics and identify conflict-related risks (Section 4.1), and ultimately make model selection decisions (Section 4.3) that successfully incorporate a complex set of fairness objectives and risks. We also provided a case study (Section 6.3) for real-world applications in order to guide practitioners in the effective use of our approach.

---

> ### Author Response · Authors · 2023-11-14
>
> > 2. The authors mentioned constrained optimization (CO) approaches in the related work. However, such a baseline is missing in the empirical evaluation. It would be better to compare CO with MaO empirically and provide more insights into the pros and cons of both approaches.
>
> Thank you for your comment regarding the lack of CO baselines in our work. Our line of thinking for not including such baselines was that CO and MO fairness approaches are designed for separate scenarios, and the best choice depends on whether an appropriate set of fairness metrics, meaningful constraints, and achievable values are known ad-hoc. Regardless, the pros and cons of MaO vs. CO, especially in the presence of multiple fairness objectives/constraints is an interesting research question.
> We would appreciate your input and thoughts on how we can follow up with experiments in this regard to compare with CO methods (we would also appreciate references to available CO methods, as the two CO papers we cited in our related work section have no public code available).

---

### Author Response · Authors · 2023-11-14
**Thank you for your feedback!**

We would like to thank all reviewers for their constructive responses. We are actively working on addressing the three main themes present in all reviews.

Firstly, regarding the novelty and technical contributions of our work, we acknowledge that while the fact that fairness metric conflicts exist is unsurprising, to the best of our knowledge we present the first approach that simultaneously optimizes different notions of fairness as conflicting objectives and thoroughly evaluates this methodology. We believe that our work opens the door to a future generation of FairML methods that acknowledges the complex landscape of FairML problems, resulting in solutions that better align with a potentially complex and nuanced set of social objectives and requirements. In addition, we believe that an exemplary use-case and theoretical framework (discussed in individual responses) for how to apply ManyFairHPO in real-world tasks would add a significant contribution.

Next, we acknowledge that an empirical comparison of ManyFairHPO with specialized bias mitigation techniques would strengthen our results, and are currently running experiments to include them.

The final piece of criticism that is present across several reviews was regarding the lack of a theoretical framework for how to select a model among different Pareto Optimal outcomes (across multiple notions of fairness), as well as commentary on how to select a candidate set of fairness metrics and identify potential conflicts. We found this criticism extremely constructive, and believe that providing such a theoretical framework will help ground our technical solution to real-world tasks, allowing practitioners to achieve better decision-making in scenarios where multiple notions of fairness are of value. In order to address this concern, we plan to provide an intended use case exemplifying the start-to-finish process from identifying relevant fairness metrics to making an informed model selection under a complex set of domain expert preferences. We believe that such an exemplary use case can be related to a more abstract framework for how to use ManyFairHPO in real-world FairML applications.

---

> ### Author Response · Authors · 2023-11-20
> **We have incorporated your comments!**
>
> We thank the reviewers for their constructive feedback and comments that help us to improve the paper significantly. Please refer to our newly submitted main text and appendix. We summarize our improvements as follows, and have also updated our responses to your individual comments:
> - We improved our methodology section (Section 4) and provided a start-to-finish framework for MaO optimization and model selection for FairML problems as presented in our new and improved Figure 1. This new framework now begins with a domain-knowledge-driven deliberation to select our performance and fairness metrics and their associated preferences, and also of identification of conflict-related risks with respect to pairs of fairness metrics (Section 4.1). The framework ends with selecting a model that effectively captures social objectives and mitigates conflict-related risk by a single objective scalarization approach (Section 4.2)
> - We provided a case study (Lawschool Admissions dataset) as an exemplary scenario in which our new framework can be applied. (Section 6.3)
> - We provided a comparison with Exponentiated Gradient Reduction (EGR) as one of state-of-the-art bias mitigation techniques. The results (Appendix A3 and Figure 8) showed that our ManyFairHPO dominates EGR in all cases except for only one case. This outcome validates the performance of our ManyFairHPO approach and encourages the community to use it for practical applications. This also paves the way for many interesting future studies.

---

### Meta-Review · Area_Chair_YQMj · 2023-12-06

**Metareview:**

This paper provides a new framework for exploring the tradeoff among various fairness criteria, and show its effectiveness in multiple datasets.

Strengths: the problem of achieving multiple notions of fairness in M is important. The paper is well-written. The authors actively responded to the reviewers' questions and suggestions and provided an updated version.

Weaknesses: reviewers generally concerned the lack of novelty, technical depth, and convincing empirical evaluations.

**Justification For Why Not Higher Score:**

Important problem, natural idea, but the contribution is not significant enough.

**Justification For Why Not Lower Score:**

N/A

---

### Decision · Program_Chairs · 2024-01-16

Reject